# Mean Aggregation in Distributed Learning is Vulnerable, even to Constrained Label Flipping

## Abstract

Machine learning systems deployed in distributed or federated environments are highly susceptible to adversarial manipulations, particularly availability attacks – rendering the trained model unavailable. Prior research in distributed ML has demonstrated such adversarial effects through the injection of gradients or data poisoning. In this study, we aim to better understand the potential of weaker (action-wise) adversaries by asking: Can availability attacks be inflicted solely through the flipping of a subset of training labels, without altering features, and under a strict flipping budget? We analyze the extent of damage caused by constrained label flipping attacks against distributed and federated learning under mean aggregation – the dominant baseline in research and production. Focusing on a distributed classification problem, (1) we propose a novel formalization of label flipping attacks on logistic regression models and derive a greedy algorithm that is provably optimal at each training step. (2) To demonstrate that severe model degradation can be induced by label flipping alone, we show that a budget of only 0.1% of labels at each training step can reduce the accuracy of the model by 6%, and that some models can perform worse than random guessing when up to 25% of labels are flipped. (3) We shed light on an interesting interplay between what the attacker gains from more *write-access* versus what they gain from more *flipping budget*. (4) We define and compare the power of targeted label flipping attack to that of an untargeted label flipping attack.

## 1  Introduction and Related Work

Machine learning systems can become prime targets for adversarial attacks. *Training-phase poisoning attacks* in particular have gained considerable attention as the widespread use of machine learning in critical applications has grown (Awasthi et al. (2017); Zhang et al. (2017); Paudice et al. (2018); Lu et al. (2022); Huang et al. (2011)). In these attacks, an adversary manipulates the training data in order to degrade or control the final trained model. Unlike evasion and backdoor attacks, poisoning requires no control over inference-time input: it suffices to manipulate part of the training set. Among such attacks, *label flipping* stands out for its simplicity: The attacker simply changes the class label of a subset of training points while leaving other aspects of the data intact. This is especially relevant in federated learning situations where features are fixed by upstream data pipelines and the set of possible labels is predefined. For example, a common scenario is when workers are asked to label predefined images, the labels being in a finite set. We address the following question in the context of a classification problem:

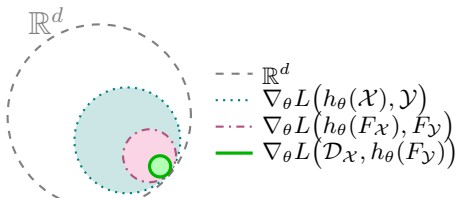

Figure 1: Images of the gradient operator on different sets. $\mathbb{R}^d$ is where an attacker can craft unrestricted gradient attacks. Given a model $h_\theta$, $\nabla_\theta L(h_\theta(\mathcal{X}), \mathcal{Y})$ is the set of possible gradients given an unrestricted data poisoning Bouaziz et al. (2024), $\nabla_\theta L(h_\theta(\mathcal{F}_\mathcal{X}), \mathcal{F}_\mathcal{Y})$ is the set of possible gradients when data poisoning is restricted to a feasible set $\mathcal{F}_\mathcal{X} \times \mathcal{F}_\mathcal{Y} \subseteq \mathcal{X} \times \mathcal{Y}$, and $\nabla_\theta L(\mathcal{D}_\mathcal{X}, h_\theta(F_\mathcal{Y}))$ is the set of possible gradients when the features are restricted to those in the dataset $\mathcal{D}_\mathcal{X}$ and the labels are chosen in the set of feasible labels.

*Can an attacker severely degrade a model using only label flips on existing data in a convex setting and under budget constraints?*

In this work, we provide a positive answer to this question. By viewing label flipping as a constrained optimization problem from the point of view of the attacker, we show that carefully selected flips can steer the aggregated gradient away from its honest direction and reduce accuracy or render training unstable - even when only 1% of the labels are altered.

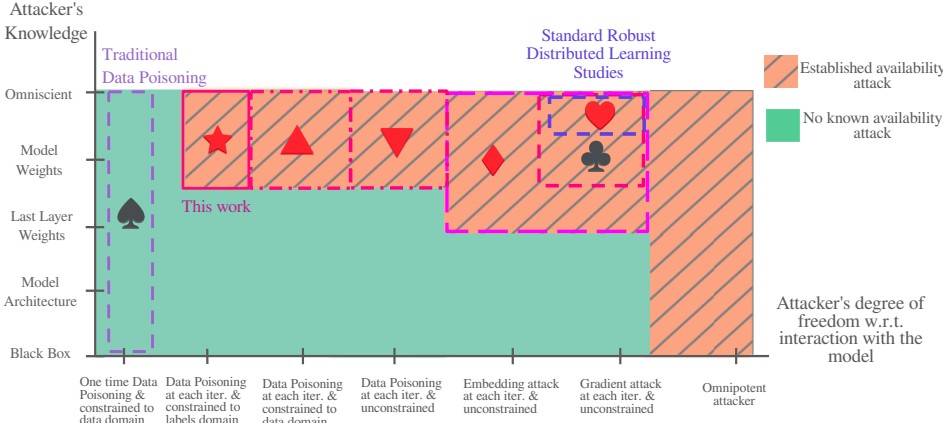

Figure 2: Territory of known availability attacks (in orange) within a domain of constraints. The closer to the origin, the more constrained is the setting for the attacker and the harder it is to realize an availability attack. ♠: Geiping et al. (2020); Zhao & Lao (2022); Ning et al. (2021); Huang et al. (2020), ♡: Blanchard et al. (2017); Baruch et al. (2019), ♣: El-Mhamdi et al. (2018), ◇ so far only in convex settings : Farhadkhani et al. (2022), △&▽ : Bouaziz et al. (2024), ★ : Our contribution in section 3 .

Previous work has investigated poisoning attacks and granted the adversary far more leverage than some realistic scenarios allow. Some require the ability to modify *both* labels and features or even overwrite entire gradients ( Bouaziz et al. (2024); Baruch et al. (2019)); others rely on injecting crafted examples into the training set ( Shafahi et al. (2018); Zhao & Lao (2022); Koh et al. (2022)). Li et al. (2022) proposes an agglomerative-clustering strategy to select vulnerable samples, assuming access to all training data and performing offline clustering. Jha et al. (2023) demonstrates that carefully chosen label poisoning can insert backdoors via trajectory-matching methods. Their objective is to create targeted misclassification when a trigger is present, not to reduce overall model accuracy. Thus, although both their work and ours manipulate labels only, the goals, threat models, and evaluation metrics differ: theirs aim to insert backdoors, whereas we study availability degradation under a tight budget and online setting. Likewise, Lavaur et al. (2024) applies random label flipping in the context of collaborative intrusion detection systems. The attack of Lavaur et al. (2024) targets domain-specific labels (e.g., DDoS) and does not formalize the attack as an optimization problem. In contrast, we aim to study an attack that is data- and model-agnostic but can be proved to be optimal at each iteration.

Yu et al. (2025) studies label-flipping attacks on GNNs under a federated learning setup. Their approach is oriented towards graph-structured data and relies on the unique properties of GNN aggregation mechanisms. In contrast, we propose a method that is formulated as a general optimization framework based on gradient alignment, making it applicable to a wider range of classification tasks. When manipulation is limited to labels alone, existing methods typically presuppose control over the vast majority of them ($\geq 85\%$) or access to the validation set, and require the training of a number of classifiers linear in the number of classes and the budget ( Liu et al. (2023a); Paudice et al. (2018)), unsuitable for online or real-time training, especially at scale.

Another recent work, Bal et al. (2025), focuses on *defense* in the binary classification setting via adversarial training (SVMs, untargeted), whereas our work formulates the *attacker's* online, budget-constrained optimization and proves per-epoch greedy optimality under explicit write-access and local budget constraints.

By contrast, our work formulates the attacker's optimization problem under a strict, per-epoch budget, with explicit write-access and local budget constraints. Our attack is online, performs per-iteration optimization in parameter space using gradient alignment, and is provably optimal per step. Furthermore, it requires only control over a small fraction of data and changes no features at all, only labels. This expands the scope of known *availability attacks* to a more constrained, yet practically damaging, threat model.

Following Bouaziz et al. (2024), Figure 2 shows label-flipping attacks on the landscape of availability attacks, situating our contribution within the literature. Meanwhile, Figure 1 illustrates how label flipping compares to more general gradient-based attacks, particularly with respect to the set of gradients achievable under increasingly restrictive conditions.

**Why mean aggregation still matters**   Mean aggregation remains the de facto baseline in both federated learning research and large-scale deployments as it is favored for its scalability, communication efficiency, and ease of implementation ( McMahan et al. (2017)). As noted in Shejwalkar et al. (2021), production cross-device FL systems with thousands to billions of clients often employ plain averaging, and empirical evidence suggests that even the most basic, non-robust FL algorithms can be surprisingly resilient in realistic, low-compromise settings. In this context, showing that a constrained, label-flipping adversary can meaningfully degrade model performance under mean aggregation is already a strong and practically relevant finding. Additional discussion is provided in Appendix 7.3.

**Contributions.**   We formalize label flipping for logistic regression as a budget-constrained optimization problem whose closed-form objective depends only on the inner products between feature vectors and a reference direction (Section 2). This formulation yields a greedy algorithm that we prove to be *optimal at each training step* (Section 3). Experiments on standard image classification benchmarks confirm the severity of the attack: altering merely 0.1% of the labels already reduces the test accuracy by 6%, a change subtle enough to bypass typical anomaly detection thresholds while still inflicting high computational, financial, and temporal costs. Also, a 25% global budget forces the model to almost random guessing (Section 4). We also discuss the trade-off between an attacker's *write-access* ($k$) and their *local budget* ($b$), showing that a wider write-access is more valuable than a larger local budget, and we compare the targeted and untargeted variants of the proposed algorithm. We then extend the framework to an arbitrary number of classes and propose a generalization of the binary label flipping attack algorithm (Sections 5 and  6). We conclude by discussing limitations of our work and future directions.

## 2   General Setting

### 2.1   Notation

The main notation used in this work is summarized in Table 1.

Although $D$ and $\alpha$ vary with $t$ (iteration dependent), we omit the epoch index whenever there is no risk of ambiguity since **we treat the attacker's problem epoch-wise**. By definition $|P| \leq b|K|$. In addition, we use $i \in D$ and $(x_i, y_i) \in D$ interchangeably, and writing $y_i \in K$ means $\{y_i$ such that $(x_i, y_i) \in K\}$.

### 2.2   Threat Model

**Rationale.**   As noted in Bouaziz et al. (2024), the fundamental difference between gradient attacks and data poisoning comes from the limited expressivity of the latter. This difference is even more pronounced for label-flipping. Following the two most relevant works to ours Farhadkhani et al. (2022); Bouaziz et al. (2024), to compare label-flipping and gradient attacks on similar grounds, we consider a threat model in which both attacks can be executed by allowing an attacker to recalculate its attack at each iteration, similarly to Algorithm 1 in Steinhardt et al. (2017).

Table 1: Notation Summary

| Notation | Description |
|---|---|
| $d$ | Dimension of the feature space. |
| $t$ | Epoch (training iteration) index. |
| $(x_n, y_n)$ | $n$-th data point, with features $x_n \in \mathbb{R}^{d+1}$ and label $y_n \in \{0, 1\}$. |
| $\alpha \in \mathbb{R}^{d+1}$ | Binary logistic regression parameter vector. |
| $W \in \mathbb{R}^{C \times (d+1)}$ | Multinomial logistic regression parameter matrix. |
| $H$ | Set of *honest* data points (labels are not flippable). |
| $K$ | Set of attacker-controlled data points (labels can be flipped). |
| $K_H$ | Honest version of $K$ before any label flips. |
| $D_H = H \cup K_H$ | Entire *honest* training dataset (unmodified). |
| $D = H \cup K$ | Entire training dataset after poisoning (some labels in $K$ may be flipped). |
| $N = \|D\| = \|D_H\|$ | Total number of data points. |
| $k = \frac{\|K\|}{\|D\|}$ | Fraction of the dataset controlled by the attacker (write-access). |
| $P \subseteq K$ | Subset of $K$ whose labels are actually flipped by the attacker. |
| $b$ | *Local per-iteration flipping budget* (proportion of $K$ that can be label-flipped). |
| $\mathbb{1}[\cdot]$ | Indicator function (returns 1 if the condition is true, 0 otherwise). |
| $\sigma(\cdot)$ | Sigmoid function: $\sigma(z) = \frac{1}{1+e^{-z}}$. |
| $k \times b$ | Corrupted fraction (Global budget) |

**General Learning Setup.** We study a classification problem where the training dataset is denoted by $\mathcal{D}_{\text{train}} = \{(x_i, y_i)\}_{i=1}^n$, with samples drawn from a distribution $\mathcal{D}$ over the input–label space $\mathcal{X} \times \mathcal{Y}$. The learner trains a neural network $h_\theta$, parameterized by $\theta \in \mathbb{R}^d$, using an iterative optimization procedure to minimize a loss function $\mathcal{L}$. The objective is to achieve low test loss on a held-out dataset $\mathcal{D}_{\text{test}}$.

We model the learning process as involving $n_w$ Gradient Generation Units (GGUs). At each iteration $t$, each unit produces a message, and these messages are aggregated using a function Agg, and the model parameters are then updated using that aggregated vector.

This abstraction allows us to represent a large spectrum of learning settings, from centralized learning to fully distributed learning and settings in between (such as federated learning).

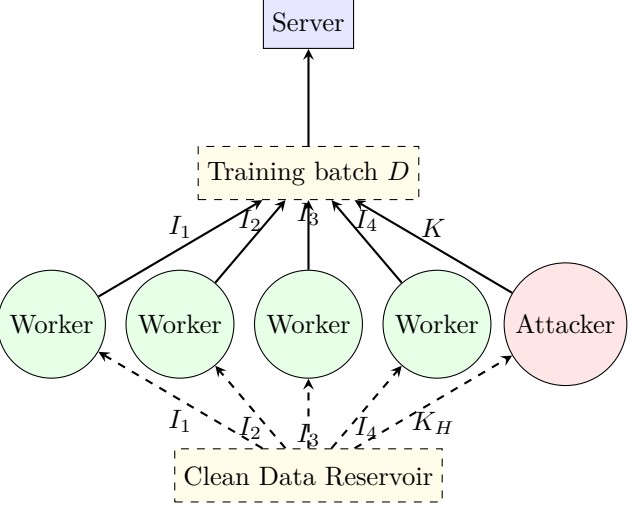

Figure 3: Illustration of the setting: Each user obtains its data from a clean reservoir. The malicious user flips up to a budget $b$ fraction of the labels in $K_H$.

**Attacker Setup.** Within this distributed framework, suppose that at every training epoch $t$, the GGUs (workers) receive data from a trusted data source and collectively transmit a batch of $N$ examples to the parameter server after processing it. A malicious worker—whom we call the attacker—is hidden among them and contributes to the final batch with a fixed fraction $k$ of data points, giving the adversary write access to that proportion $k$ of the data. Let $K_H$ denote the clean examples received by the compromised worker. Before forwarding its message, the attacker flips the labels of at most a fraction $b$ of these examples (its local budget), creating the manipulated set $K$.

The server then trains on $K \cup H$ where $H = \cup_j I_j$ is the unaltered data from the remaining honest workers (each contributing $I_j$). Therefore, no more than a $k \times b$ fraction of the epoch's batch (the attacker's global budget) is corrupted. Throughout this work, we consider that the server's aggregator Agg uses mean

aggregation of gradients. Figure 3 illustrates this setup. The attacker's control over data points in $K$ is strictly on their labels; feature vectors remain unaltered. Prior work on gradient-based attacks Bouaziz et al. (2024); Baruch et al. (2019) assumes that the adversary is omniscient, therefore, we allow the attacker to be *omniscient*: they have full *read-access to the model parameters* at every epoch. Further details on the threat model can be found in section 7.4.

**On the weak adversary** Throughout the paper *weaker adversary* refers strictly to the action space: the attacker may only flip labels (under a strict budget, no feature modification, no arbitrary gradient injection). We also note that our formulation assumes a Horizontal Federated Learning setting where the attacker knows the ground-truth labels of their local data. We adopt an omniscient adversary (read access to parameters each round) to align with prevalent threat models in gradient-based poisoning/Byzantine-robust literature Chen et al. (2017); El-Mhamdi et al. (2018); Xie et al. (2018) and to enable a clean comparison: how close can a labels-only attacker get to classical availability attacks? For the more realistic limited-knowledge case, Appendix 7.4 shows how an attacker can conceptually train a local surrogate on its accessible subset $K_H$ and apply the same greedy selection there.

### 2.3 Label Flipping as a Constrained Optimization Problem

At epoch $t$, the server would normally evaluate the empirical loss on the *honest* batch $L_{D_H}(\alpha_t) = \frac{1}{N}\sum_{i \in H \cup K_H}\ell_i(\alpha_t)$, and update the model with the corresponding gradient $\nabla L_{D_H}(\alpha_t)$. However, once the man-in-the-middle adversary flips some labels, the server instead observes the *poisoned* batch $D = H \cup K$ and the loss $L_D(\alpha_t) = \frac{1}{N}\sum_{i \in H \cup K}\ell_i(\alpha_t)$, where $\ell_i$ is the per-sample cross-entropy loss defined in section 3.

The attacker decides which labels in $K$ to flip so that the poisoned gradient $\nabla L_D(\alpha_t)$ is as *misaligned* as possible with a chosen direction $\Delta$. We distinguish two goals:

$$
\Delta = \begin{cases} -\nabla L_{D_H}(\alpha_t), & \text{untargeted attack}, \\ -\left(\alpha^{\text{Target}} - \alpha_t\right), & \text{targeted attack}. \end{cases}
$$

The untargeted adversary tries to deviate from the honest gradient, whereas the targeted adversary steers the update toward a pre-selected parameter vector $\alpha^{\text{Target}}$. Figure 4 illustrates how the targeted variant uses the vector $-(\alpha^{\text{Target}} - \alpha_t)$ to bias each gradient step toward $\alpha^{\text{Target}}$. The choice of the target parameter is explained in Appendix A.

Formally, at each epoch $t$, the attacker solves:

$$
\underset{\{y_i^{(D)}\}_{i \in K}}{\arg\min} \left\langle -\nabla L_D(\alpha_t),\ \Delta \right\rangle \tag{1}
$$

$$
\text{s.t.} \underbrace{\sum_{i \in K}\mathbf{1}\left[y_i^{(D)} \neq y_i^{(D_H)}\right] \leq b\,|K|}_{\text{(Budget constraint)}} \tag{BC}
$$

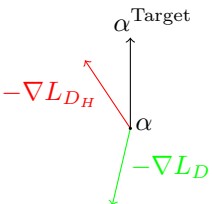

Figure 4: Desired training step direction of honest workers (red), and that of the targeted (black) and untargeted (green) attackers.

where $y_i^{(D_H)}$ are the clean labels received from the compromised worker and $y_i^{(D)}$ are the (possibly flipped) labels that the attacker forwards to the server. The budget constraint equation BC enforces the local budget $b$: at most a fraction $b$ of the $|K|$ labels under the attacker's control can be flipped.

## 3 Attack Method in the Binary Setting

In this section, we focus on a binary logistic regression classifier. For a sample $(x_n, y_n) \in \mathbb{R}^{d+1} \times \{0, 1\}$, the *cross-entropy* loss is $l_n(\alpha) = -\left[y_n \log\left(\sigma(\alpha^\top x_n)\right) + (1 - y_n)\log\left(1 - \sigma(\alpha^\top x_n)\right)\right]$.

### 3.1 Problem Formulation

At each epoch the server has a parameter vector $\alpha \in \mathbb{R}^{d+1}$ (the last coordinate is the bias) and receives a mini-batch $D = H \cup K$ of size $N$, where $H$ are honest examples and $K$ are examples under the attacker's control. For logistic regression, the batch gradient is

$$\nabla L_D(\alpha) \; = \; \frac{1}{N} \sum_{n=1}^{N} \big(\sigma(\alpha^\top x_n) - y_n\big) x_n \; = \; \frac{|H|}{N} \nabla L_H(\alpha) \; + \; \frac{|K|}{N} \nabla L_K(\alpha), \tag{2}$$

with an analogous decomposition for any subset of the data.

Recall from equation 1 that the adversary chooses the labels in $K$ so as to *anti-align* the poisoned gradient with a reference direction $\Delta$ (either the honest gradient or the displacement toward a target model). Because the honest labels are immutable, only the term $\nabla L_K(\alpha)$ matters for the optimization.

Using $\langle \Delta, -\nabla L_K(\alpha) \rangle |K| \; = \; \sum_{i \in K} \langle \Delta, x_i \rangle \big(y_i - \sigma(\alpha^\top x_i)\big)$, the problem equation 1 simplifies to

$$\underset{\{y_i^{(D)}\}_{i \in K}}{\arg\min} \sum_{i \in K} \langle \Delta, x_i \rangle y_i^{(D)} \quad \text{s.t.} \quad equation\ BC, \tag{3}$$

Where $y_i^{(D)} \in K$ means the label of data point $(x_i, y_i) \in K$.

The objective reveals a clear strategy: to maximise gradient distortion, flip those examples whose feature vectors $x_k$ have the *largest negative* inner product with $\Delta$, i.e. those most misaligned with the desired update direction.

### 3.2 A Greedy Label-Flipping Algorithm for Binary Classification

Based on the previous formulation, we now provide an explicit algorithm for the attacker's label flipping strategy which is *provably optimal at each epoch*. For each attacker-controlled point $(x_i, y_i) \in K$, consider the scalar product $s_i = \langle \Delta, x_i \rangle$. Notice that giving a label of 1 to the points whose $s$ is negative, and a label of 0 to the others will give the minimum of the objective function at the current iteration.

If only a fraction $b$ of the points in $K$ can be flipped, the attacker should focus flips on those $x_i$ that yield *the most misaligned* values $s_i$ (those that have the greatest magnitude). Concretely, define $p = \lfloor b \cdot |K| \rfloor$. Then: 1. Identify the $p$ points whose $s_i$ is *smallest*. 2. Flip each of those $p$ points to label 1 if $s_i < 0$, or 0 if $s_i \geq 0$.

If the attacker is allowed to flip *all* data points in $K_H$, then the strategy is applied to all its points. Algorithm 3 (Appendix A), whose optimality at each epoch is proven in appendix B.1, describes the label flipping strategy. Details on the training algorithm, the hyperparameters, and the target model used can be found in the appendix.

## 4 Binary-Classification Experiments

### 4.1 Overall Attack Impact

Experiments on MNIST show that by flipping labels with a global budget $k \times b \leq 25\%$ of the data at each epoch, the attacker can perform an availability attack and keep the model at a random level. Even a global budget of 0.1% reduces the accuracy by around 6%. The exact effect depends on the dataset and model, but the attack tends to be more effective on CIFAR-10, CIFAR-100, and when using an MLP compared to logistic regression.

Another observation is a monotonic trend: increasing $k$ or $b$ strengthens the attacker's ability to degrade performance or push the parameters toward a desired target. We can also see that for the given $b$ the accuracy decreases as a function of $k$, however, it is still inherently limited due to the nature of the task and the form

of the loss: It is (up to a constant) *a linear combination of N feature vectors with binary weights* that limits the number of directions we can use during loss minimization.

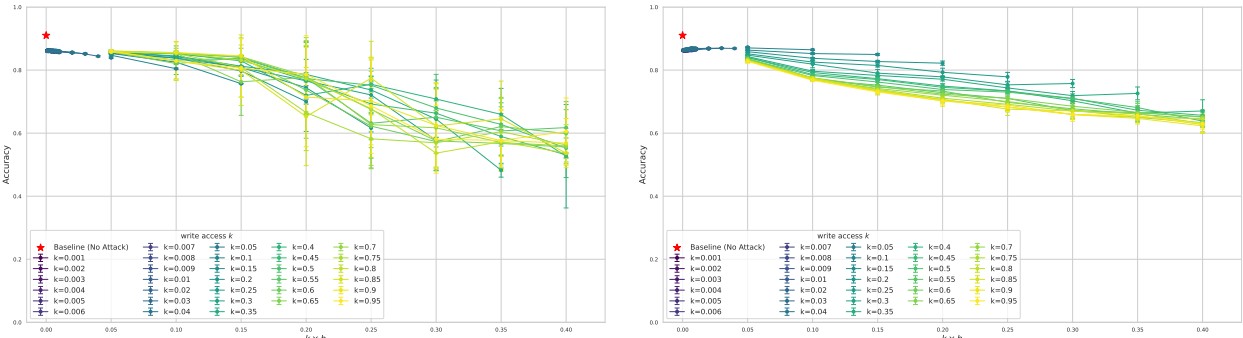

Figure 5: Binary classification: Test accuracy as a function of the global budget under untargeted (left) and targeted (right) attacks, for Mnist (classes 0 and 1).

## 4.2 Untargeted vs Targeted Attacks: Impact and Variance.

The histogram in Figure 6 and the heat map in Figure 7 provide an interesting perspective on how untargeted and targeted label flipping attacks compare in a binary classification setting. At low levels of corruption (for example, $k < 0.1$), both attacks produce a similarly low variance in final accuracy. This indicates that a small amount of label flipping–whether targeted or untargeted–does not drastically affect the stability of model training. However, as $k$ increases beyond about 0.1, the variance in accuracy begins to grow exponentially, suggesting that the model performance becomes increasingly sensitive to label corruption.

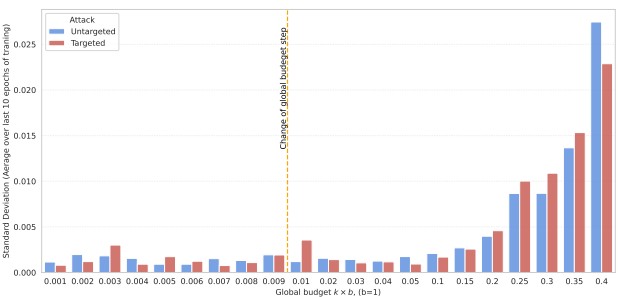

Figure 6: Standard deviation of the final accuracy as a function of $k$ ($b = 1$). The vertical line marks the point where the step size of the $x$-axis changes.

## 4.3 Write-access vs. Local budget trade-off

When looking more closely at the interaction between $k$ (write-access) and $b$ (flipping budget), the heat map in Figure 7 reveals subtle distinctions. Specifically, when $k \lesssim 0.2$, there is very little difference between untargeted and targeted attacks in terms of their overall impact. This similarity makes intuitive sense: at moderate or low corruption rates, flipping is not pervasive enough –whether untargeted or targeted–to cause consistently divergent behaviors in how the model updates its predictions. However, once $k \gtrsim 0.2$, the nature of the attack begins to matter more since untargeted attacks become more efficient, due to the optimality of untargeted attacks.

Nevertheless, the scale of these differences, on the order of 0.2, is not large enough to be of major practical significance in typical real-world use cases. In many binary classification tasks, the difference in mean accuracy (and variance) induced by untargeted versus targeted label flipping is relatively modest.

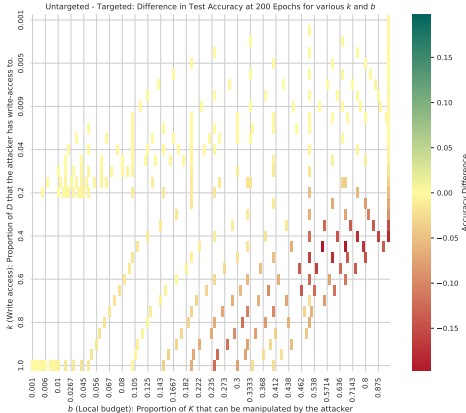

Figure 7: Heatmap of the difference of test accuracies between an untargeted attack and a targeted attack.

From a robustness standpoint, this suggests that the primary concern should be the overall fraction of corrupted labels rather than the specific pattern of flipping.

Recall that the attacker is omniscient and that they have read-users, however, they are limited in their write-access by $k$ and by total flipping proportion $k \times b$, *is it better to increase b and decr*

Figure 8 shows the test accuracy for different combinations of $k$ and $b$. We see that the greater $k$ is, the more effective the attack, and for small $k$ values, b has no impact on the test accuracy. From this, we infer that it is more impactful from the point of view of the attacker to have wide write-access, so the priority is for $k$ before $b$ for a given total flipping proportion $k \times b$. This can be understood by the fact that at each iteration the gradient of the loss as formulated in equation 2 is a weighted linear combination of the feature vectors, and these weights are linear in the labels which are discrete, which limits the space of gradients to finite set of vectors. Hence, having a greater *write-access* provides a richer space of gradients.

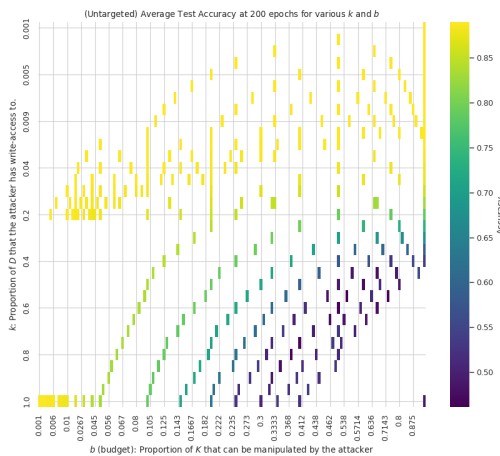

Figure 8: Heatmap of test accuracy as a function of $k$ and $b$ at 200 epochs for the untargeted attack.

## 5 Extending the Attack to Multi-class tasks

In this section, we extend our setting to analyze how label flipping affects model performance in the multi-class classification problem. Consider that we have $C$ classes and again $N$ data points and define a matrix $\mathbf{T}$ that encodes the classes of data points in $D$. We use one hot encoding to encode classes and cross entropy as the loss. For $c \in [1, C]$ and $n \in [1, N]$, $\mathbf{T}_{cn}$ corresponds to whether the $n$-th data point has label $c$ or not: $\mathbf{T}_{cn} = 1$ if $y^{(n)} = c$ and $\mathbf{T}_{cn} = 0$ otherwise. And let $W \in \mathbb{R}^{C \times (d+1)}$ be the matrix of parameters of the logistic regression model.

The cross entropy loss can be written as:

$$L_D(W) = -\sum_{n=1}^{N} \sum_{c=1}^{C} \mathbf{T}_{cn} \log p(\mathbf{T}_{cn} = 1 | \mathbf{x}^{(n)}, W) = -\sum_{n=1}^{N} \sum_{c=1}^{C} \mathbf{T}_{cn} \log \operatorname{softmax}(\mathbf{W}\mathbf{x}^{(n)})_c$$

Following the same idea as in the binary setting, the attacker wants to flip labels to control the gradient. Let $\nabla_j L_D$ be the gradient of the loss on dataset $D$ with respect to the $j^{th}$ row of $W$.

$$\nabla_j L_D = \frac{\partial L_D}{\partial W_{j,\bullet}} = -\sum_{n=1}^{N} \sum_{c=1}^{C} \mathbf{T}_{cn} \frac{1}{\operatorname{softmax}(\mathbf{W}\mathbf{x}^{(n)})_c} \frac{\partial}{\partial W_{j,\bullet}} \operatorname{softmax}(\mathbf{W}\mathbf{x}^{(n)})_c$$

$$= -\sum_{n=1}^{N} \sum_{c=1}^{C} \mathbf{T}_{cn} \left( \mathbb{1}[c=j] - \operatorname{softmax}(\mathbf{W}\mathbf{x}^{(n)})_j \right) \mathbf{x}^{(n)}$$

Where $\mathbb{1}[\cdot]$ is the indicator function.

For $j \in [1, C]$, let $\Delta_j = \begin{cases} -\nabla_j L_{D_H}, & \text{(untargeted attack)} \\ -\left(W^{\text{Target}} - W_t\right), & \text{(targeted attack)} \end{cases}$ where $W^{\text{Target}}$ is a target model and $W_t$ the model at the iteration $t$. The optimization problem of the attacker at each epoch is to minimize the Frobenius inner product:

$$\underset{T \in \{0,1\}^{C \times N}}{\arg\min} \quad \sum_{j=1}^{C} F_j(T)$$

$$\text{s.t.} \quad \begin{cases} F_j = \langle \nabla_j L_D, -\Delta_j \rangle = \sum_{n=1}^{N} \sum_{c=1}^{C} T_{cn} \left( \mathbb{1}_{\{c=j\}} - \operatorname{softmax}(Wx^{(n)})_j \right) \langle x^{(n)}, \Delta_j \rangle, \\ \sum_{c=1}^{C} T_{cn} = 1 \quad (\forall n), \\ equation\ BC \end{cases}$$

Let $Z_{cn} = \sum_{j=1}^{C} \langle \mathbf{x}^{(n)}, \left( \mathbb{1}[c = j] - \text{softmax}(\mathbf{W}\,\mathbf{x}^{(n)})_j \right) \Delta_j \rangle$. The constraints impose that $\sum_{c=1}^{C} \mathbf{T}_{cn} Z_{cn}$ is in reality just one term. Therefore, for $n \in K$, take $c^{(n^*)}$ as the index of the minimum of $(Z_{cn})_{c \in C}$ and assign $T_{cn} = 1$ if $c = c^{(n^*)}$ and 0 otherwise, starting with the $Z_{cn}$s that yield the least until we run out of budget. Taking such labels that minimize $Z_{cn}$ for every $n$ ensures that the attacked gradient is minimal across all other possible label choices. Meaning that the algorithm is per-epoch optimal. Algorithm 1 details the procedure.

---

**Algorithm 1** Per-epoch greedy label selection (multi-class)

---

**Require:** Attacker set $K = \{x^{(n)}\}_{n \in K}$, current weight matrix $W_t \in \mathbb{R}^{C \times (d+1)}$, vectors $\Delta_j$ for $j = 1, \ldots, C$, local budget fraction $b \in (0, 1]$
**Ensure:** Label assignment indicators $T \in \{0, 1\}^{C \times N}$ for $n \in K$
1: $p \leftarrow \lfloor b \cdot |K| \rfloor$
2: **for** each $n \in K$ **do**
3:     **for** each class $c \in \{1, \ldots, C\}$ **do**
4:         compute
$$Z_{cn} \leftarrow \sum_{j=1}^{C} \langle x^{(n)}, \; (\mathbb{1}[c = j] - \text{softmax}(W_t x^{(n)})_j)\, \Delta_j \rangle$$
5:     **end for**
6:     $c^\star(n) \leftarrow \arg\min_c Z_{cn}$
7:     $Z_{\min}(n) \leftarrow Z_{c^\star(n),n}$
8: **end for**
9: Sort indices $n \in K$ by ascending $Z_{\min}(n)$ and select the $p$ smallest indices $S$.
10: **for** each $n \in S$ **do**
11:     Set $T_{c^\star(n),n} \leftarrow 1$ and $T_{c,n} \leftarrow 0$ for $c \neq c^\star(n)$
12: **end for**
13: **Return** $T$

---

## 6 Multi-Class Experimental Results

**Experimental setup.** All experiments report means $\pm$ standard variation over 6 independent runs. Datasets: MNIST, CIFAR-10, CIFAR-100 (details in App. A). Models: logistic regression (main) and a 2-layer MLP. Data splits are iid across workers. Optimizers: SGD (main) and Adam (ablation, App. A). Metrics: test accuracy and F1-score. All the figures are shown for Mnist dataset (where the attack has the least impact accross the datasets tested). Results for the other datasets and the vanilla neural network is in Appendix C.

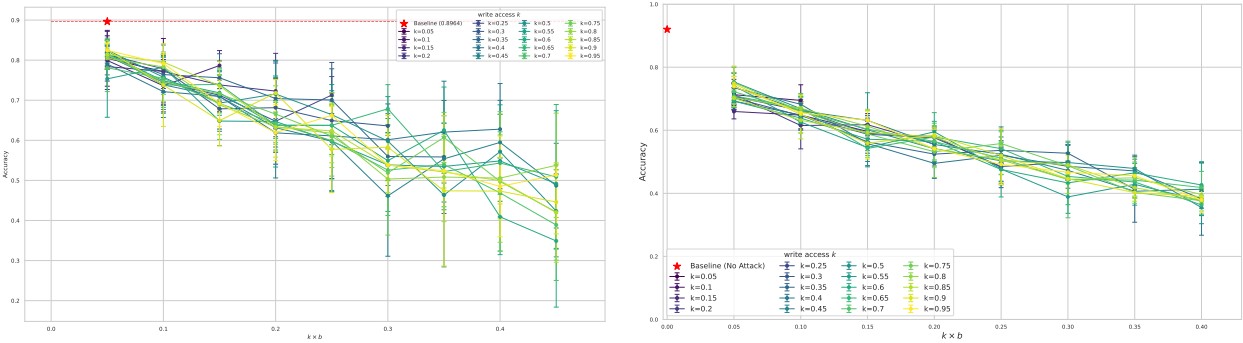

Figure 9: Test accuracy vs. global budget under untargeted (left) and targeted (right) attacks, for Mnist10 dataset (unnormalized).

Figure 9 shows the attack results. Each plotted point reports the average test accuracy as a function of the fraction of modified points, and is an average over 6 runs. It shows that for just a global budget of 5% of the dataset, the attacks dropped the test accuracy by $10 - 19\%$. Moreover, Figure 10 shows that the variance of an untargeted attack is greater than that of a targeted attack, confirming the idea that untargeted attacks are more chaotic since they are not directed whereas targeted attacks are more guided and hence less erratic. Also, comparing Figure 10 with Figure 6 shows that the magnitude of the accuracy's variance under attack grows with the number of classes, going from around 0.02 to 0.1 as the number of classes goes from 2 to 10. Which suggests that effect of the proposed attack on the accuracy's variance grows with the number of classes.

**Detectability and data normalization.** Many practical FL pipelines normalize or scale features prior to training. We empirically observe that under min-max normalization the attacker's per-client gradient norm is nearly indistinguishable from honest clients (see Table 3 in Appendix E). Consequently, defenses that clip updates based solely on gradient norm thresholds are ineffective in normalized settings. When inputs are unnormalized, the gradient-norm disparity becomes large and simple norm-based defenses regain efficacy. That's a limitation of the attacker since he can't control the features.

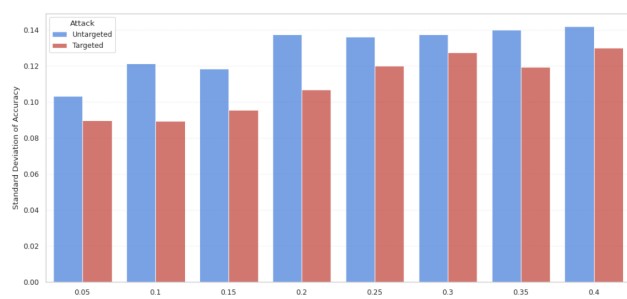

Figure 10: Standard deviation of test accuracy vs global budget (b fixed =1) for the multi-class classification setting.

## 7 Discussion

### 7.1 How practical is the attack?

In addition to our formal setup, allowing comparison between gradient attacks and label flipping, we note several realistic settings where label-only corruption can occur in production systems while possibly having acces to knowledge about the model. This can be the case in, *e.g.*,

- compromised labeling tools: attackers who can modify annotation UIs, metadata, or labeling scripts can change labels stored in databases while leaving raw sensor data untouched,

- insider at a data provider: an employee with access to label fields (but not feature pipelines) can flip labels locally,

- crowdsourced annotation manipulation: attackers controlling a subset of crowd workers or poisoning weakly-supervised labeling heuristics can flip a small fraction of labels across contributors, as well as in,

- collaborative learning situations where participants are provided various forms of "a global model" in exchange for their participation with data as in, *e.g.* Capitaine et al. (2024); Ananthakrishnan et al. (2024).

### 7.2 Why Logistic Regression (before deeper models)

There are two reasons why we adopt logistic regression as a first-class test bed before moving to deeper models as we did: *(i) Analytic transparency.* The loss is convex and its gradient admits a simple closed form, allowing us to derive our greedy attacks. *(ii) Gateway to richer models.* Any classifier whose final layer is linear followed by a sigmoid/softmax (e.g. multilayer networks, convolutional nets) reduces to logistic regression if we treat the penultimate activations as "features." Consequently, at each training step our attack should apply to those models by operating on the last-layer logits.[1] This raises the question: Would such an attack

---

[1]Formally, let $f_\theta(X) = \sigma(W h_\phi(X))$ with inner network $h_\phi$ fixed during one optimization step. Setting $\tilde{X} = h_\phi(X)$ recasts the update as a logistic regression in $(\tilde{X}, y)$ with parameter $W$.

still be greedy for any classifier? One path to performing a label-flipping attack on a broader class of models could be the transition from logistic regression to neural networks. We perform experiments on a 2-layer vanilla MLP and a ViT using the method described above, the results can be found in Appendix C, and show that the attack is remarkably efficient even for these deeper models. Another approach is to find a globally optimal label flipping attack algorithm. We assumed that the server was aggregating the gradients using the mean, and a future work could discuss similar attacks on settings with different aggregation methods presented in  Baruch et al. (2019); El-Mhamdi et al. (2018); Blanchard et al. (2017).

### 7.3   Why mean aggregation still matters

Our focus on mean aggregation is intentional: it establishes a baseline benchmark for the impact of label-flipping attacks. If such an attack fails here, it is unlikely to succeed against more sophisticated robust aggregators; if it succeeds, it provides a clear lower bound that any robust method should surpass. Moreover, our attack design is specifically derived for the mean operator, and different aggregation rules would require different attack formulations. In real-world deployments, where FedAvg is still prevalent, understanding vulnerabilities under mean aggregation is crucial for designing defenses.

Further, our results are meant as an opening for constrained label flipping attacks. By demonstrating that the constrained label-flipping attack is feasible against the most common aggregation rule, we establish the case for extending our methodology to robust aggregation methods as a natural next step. We also note that some defenses like norm clipping (which are shown very efficient in practice in combination with mean aggregation ( Sun et al. (2019)) are inherently blind against our approach, since the attack operates in the space of valid gradient updates derived from valid gradient vectors generated by plausible label flips, where feature inputs are untouched. Since every flipped label still produces a legitimate gradient, clipping based solely on gradient norm is unable to distinguish or suppress malicious updates that conform to honest-looking magnitudes-making our attack inherently robust against such defenses. This further values studying mean aggregation as a baseline before moving to more complex settings.

### 7.4   What if the attacker lacks omniscience

In the case where the attacker lacks full knowledge, for example consider a scenario where the attacker knows the architecture of the server's model but not its current parameters. Then the attacker can train a local surrogate model using only the data available to them (i.e., the subset they control). This local model may not perfectly match the server's model, but it can still approximate the decision boundary or the gradient dynamics well enough to inform a useful attack as done in, *e.g.* El-Mhamdi et al. (2018); Baruch et al. (2019) or anticipate the dynamics in one-time attack as in Jha et al. (2023).

Specifically, the attacker can simulate the label-flipping attack on their local model concurrently with the server's training. By observing how different label flips affect the gradient direction or training loss on their surrogate, the attacker can estimate which flips are most likely to have the desired effect when sent to the server. This strategy may reduce the attack's effectiveness but remain viable.

## 8   Conclusion

In this study, we demonstrate that a purely *label-flipping* adversary - constrained by a strict budget and guided only by a greedy rule - can launch an availability attack. By introducing an intuitive, budget-aware objective, we reveal a vulnerability previously believed to require gradient overwrites or feature-level poisoning. Both targeted and untargeted flips destabilize training and reduce test accuracy. Our experiments on several benchmarks confirm the potency of the attack relative to the state-of-the-art baselines. These findings establish a foundation for stronger defenses and, more broadly, a deeper understanding of security in federated and distributed learning. Interesting follow-up avenues could include (i) generalizing the attack to deeper networks and non-mean aggregators (e.g. medians); (ii) searching for globally optimal flipping attacks; and (iii) devising practical defenses tailored to the proposed threat model. We believe that our work establishes a solid foundation for future advancements in secure and robust federated learning.

## Broader Impact Statement

In this paper, we improve the community's understanding of the capabilities of label flipping in damaging a the training of a machine learning model. While doing so requires the study and the development of train-phase adversarial attacks, we believe that these attacks serve a mostly pedagogical role and help alert on the potential damage done by an omniscient attacker, previously thought to be able to do harm only when allowed to send gradients in a distributed/federated learning setup.

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

## Appendix

This appendix is organised as follows, section A details the datasets, data splits, target models, and training protocol, and provides the end-to-end pseudocode used in our experiments. In Section D, we examine robustness under alternative metrics and optimizers: the F1 score and test accuracy degrade monotonically with the corrupted fraction $k \times b$, early stopping amplifies this effect, and Adam exhibits larger drops than SGD. Section B.1 presents the proof that our label-selection rule is greedy-optimal at each epoch. We then justify the focus on mean aggregation, arguing it remains a strong and practically relevant baseline and that our labels-only attack operates within valid gradient magnitudes, limiting the effectiveness of norm-based defenses. Subsequent sections discuss data distribution assumptions—our attack is distribution-agnostic and heterogeneity can further hinder robust aggregators—and clarify the threat model: we analyze an omniscient adversary for comparability and outline how a surrogate-based attacker can operate with partial knowledge.

## A    Dataset and Experimental Setup

**Dataset.** We perform all experiments on the MNIST ( LeCun et al. (2010)) handwritten-digit corpus, CIFAR-10 and CIFAR-100. Each image was flattened for the logistic regression model and we maintained the original pixel intensity scale without additional normalization for the main experiments. We later discuss the effect of normalization in Appendix F.

| Name | # Features | # Train/test | Target model |
|---|---|---|---|
| MNIST (0 vs 1) | $28 \times 28$ | 6903 / 7877 | Fully flipped ($0 \leftrightarrow 1$) |
| CIFAR10 (airplane vs automobile) | $3 \times 32 \times 32$ | 5000 / 1000 | Fully flipped (airplane $\leftrightarrow$ automobile) |
| MNIST (10-class) | $28 \times 28$ | 60000 / 10000 | Cyclic shift $y \mapsto (y+1) \bmod 10$ |
| CIFAR10 (10-class) | $3 \times 32 \times 32$ | 50000 / 10000 | Cyclic shift $y \mapsto (y+1) \bmod 10$ |
| CIFAR100 (100-class) | $3 \times 32 \times 32$ | 50000 / 10000 | Cyclic shift $y \mapsto (y+1) \bmod 100$ |

Table 2: Datasets and target models.

**Choice of target parameter vector.** For the targeted label-flip attack we adopt a deterministic rule for selecting the adversarial target label. Specifically, we use a cyclic permutation of class indices: for any original label $y_i \in \{0, \ldots, 9\}$, the attacker sets

$$y_i' \leftarrow (y_i + 1) \bmod 10.$$

To obtain the target parameter vector/matrix $\alpha^{\text{Target}}/W^{\text{Target}}$ used in the optimization, we train an honest model on the previously *flipped* dataset. The final parameters of that trained model constitute $\alpha^{\text{Target}}/W^{\text{Target}}$; the attacker then attempts to steer the server model toward these parameters via the targeted attack.

**Implementation Details.** We train logistic regression classifiers for 200 epochs using mini-batch SGD with a batch size of 64 and a learning rate of 0.001, using cross-entropy loss. At each epoch, the omniscient attacker observes the current model parameters and gradients, then flips the labels of a randomly assigned subset $K$ from the clean data pool accordingly. All results are averaged over six independent runs with different random seeds. The global training algorithm can be found below (Algorithm 2.

---

**Algorithm 2** Full Training with Label Flipping Attack

---

**Require:** Clean dataset $D$, model $M$, total epochs $E$, budgets $k$ and $b$, and functions:
- `getSubset`: retrieves the attacker's subset randomly from $D$, of size $k \times |D|$.
- `selectFlip`: determines which labels to flip, and flips accordingly using Algorithms 3 and 1.
- `trainStep`: performs one training iteration.

**Ensure:** Poison-trained model $M$
1: Initialize model $M$
2: **for** epoch $\leftarrow 1$ **to** $E$ **do**
3: $\quad K_H \leftarrow$ `getSubset`$(D, k)$
4: $\quad K \leftarrow$ `selectFlip`$(D, K, M, b)$
5: $\quad M \leftarrow$ `trainStep`$(M, (D \setminus K_H) \cup K)$ {Train on poisoned dataset}
6: $\quad D \leftarrow (D \setminus K) \cup K_H$ {Clean D for next iterations}
7: **end for**
8: **return** $M$

---

**The greedy label flipping attack in the binary setting** We present Algorithm 3, the greedy label flipping attack algorithm used for binary classification.

---

**Algorithm 3** The greedy label flipping attack for binary classification.

---

**Require:** Attacker set $K = \{(x_i, y_i)\}$ ; budget $b \in (0, 1)$; honest gradient $\nabla L_{D_H}(\alpha)$ at current epoch $t$.
$\quad p \leftarrow \lfloor b \cdot |K| \rfloor$.
$$\Delta \leftarrow \begin{cases} -\nabla L_{D_H}(\alpha), & \text{(untargeted attack)} \\ -(\alpha^{\text{Target}} - \alpha_t), & \text{(targeted attack)} \end{cases}$$
$\quad$ **for** each $i \in K$ **do**
$\quad\quad s_i \leftarrow \langle \Delta, x_i \rangle$.
$\quad$ **end for**
$\quad$ Find the $p$ indices $i$ with the smallest $s_i$.
$\quad$ **for** each selected index $i$ **do**
$\quad\quad$ **if** $s_i < 0$ **then**
$\quad\quad\quad y_i \leftarrow 1$.
$\quad\quad$ **else**
$\quad\quad\quad y_i \leftarrow 0$.
$\quad\quad$ **end if**
$\quad$ **end for**

---

# B   On the Greediness of the Proposed Algorithms

## B.1   Proof of the greediness

We now show that Algorithm 3 label flips *provably minimize* the attacker's objective *at each epoch*.

First, recall the following lemma:

**Lemma B.1** (Rearrangement Inequality)**.** *For any real numbers $x_1 \leq x_2 \leq \cdots \leq x_n$ and $y_1 \leq y_2 \leq \cdots \leq y_n$, and for every permutation $\sigma$ of $\{1, 2, \ldots, n\}$,*

$$x_1 y_n + x_2 y_{n-1} + \cdots + x_n y_1 \; \leq \; \sum_{i=1}^{n} x_i \, y_{\sigma(i)} \; \leq \; x_1 y_1 + x_2 y_2 + \cdots + x_n y_n.$$

**Proof.** Recall that for each attacker-controlled point $i \in K$, we define

$$s_i \; = \; \langle \Delta, \, x_i \rangle, \quad \text{where} \quad \Delta = -\nabla L_{D_H}(\alpha).$$

The attacker's task is to solve

$$\min_{\{y_i\}_{i \in K}} \sum_{i \in K} s_i \, y_i \quad \text{subject to} \quad \sum_{i \in K} \mathbb{1}[y_i^{(D)} \neq y_i^{(D_H)}] \; \leq \; b\,|K|,$$

where $y_i \in \{0, 1\}$ are the (possibly flipped) labels under budget $b$.

Let $m = |K|$, and assume $s_{(1)} \leq s_{(2)} \leq \cdots \leq s_{(m)}$ is the ascending order of the scalar products. Then we can re-index the labels as $\mathbf{y} = (y_{(1)}, y_{(2)}, \ldots, y_{(m)})$ so that $y_{(j)}$ pairs with $s_{(j)}$.

By the Rearrangement Inequality (Lemma B.1), for two sorted sequences $\{x_1 \leq \cdots \leq x_m\}$ and $\{y_1 \leq \cdots \leq y_m\}$, the minimum of $\sum_{j=1}^{m} x_j \, y_{\sigma(j)}$ over all permutations $\sigma$ occurs when the largest $x_j$ pairs with the smallest $y_j$, and vice versa. In our case, $y_i \in \{0, 1\}$. Thus, to minimize $\sum_{i \in K} s_i y_i$, we should assign $y_i = 1$ to the smallest $s_i$ (those that are negative) and $y_i = 0$ to the largest $s_i$ (nonnegative)—exactly as in Algorithm 3. Constrained by $\lfloor b|K|\rfloor$ total flips, the attacker picks the $\lfloor b|K|\rfloor$ smallest $s_i$ to flip to 1 when $s_i < 0$, or to 0 if $s_i \geq 0$. This guarantees local optimality at each epoch.

## B.2   On the optimality of the approach

While Lemma B.1 proves that our greedy label-selection rule is optimal within a single training epoch, this guarantee does not extend automatically to the full training trajectory. Temporal coupling of successive updates can in principle produce cancellations: a flip that maximally misaligns the gradient at step t might later be undone by the dynamics of the optimizer or by subsequent updates. Constructing a globally optimal sequence of flips therefore requires solving a combinatorial, horizon-wide optimization problem (or equivalently optimizing over label sequences), which would be computationally intractable in realistic settings and would impose substantial computational overhead compared to our per-epoch rule. Therefore, conceptually, non-greedy strategies could sometimes outperform the per-epoch greedy policy, but any practical implementation must trade off attack effectiveness against the high computational requirements of such methods.

Also, even if an individual poisoned gradient is later partially reversed, that intermediate misdirection still disrupts the optimizer's path and can significantly slow convergence – a cost that translates directly into extra training iterations (and thus compute and monetary cost) for large models.

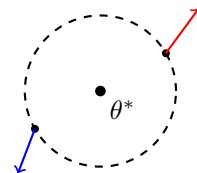

Figure 11: Illustration of temporal oscillation: two successive poisoned gradients can point in approximately opposite directions relative to the $\theta^*$. Even when later updates partially cancel earlier misdirections, the transient deviations still perturb the optimizer's trajectory and can increase training cost or degrade final performance.

### B.3   On the data distribution

Our experiments used a uniform i.i.d. data split across workers. All clients sampled from the same distribution and contributed equal-sized batches. However, our attack is distribution-agnostic: it does not assume i.i.d. data and optimizes flips online per iteration. In fact, heterogeneity generally makes many robust aggregation rules less reliable; non-IID gradients enlarge natural variability, which both weakens anomaly tests and can cause robust rules (e.g., trimming) to underperform FedAvg in certain regimes ( Liu et al. (2023b)).

As attacks are easier in heterogeneous settings, and harder in homogeneous ones (Peng et al. (2025); Karimireddy et al. (2020)), our work highlights the power of label flipping even on i.i.d. (and thus less vulnerable) situations.

## C   Experiments on more datasets and extension to deep models

Across MNIST10, CIFAR10, and CIFAR100, for both logistic regression and a 2-layer MLP, we observe a consistent pattern: our greedy label-flipping attack remains effective even when the model becomes deeper or the dataset becomes more complex or there are more classes.

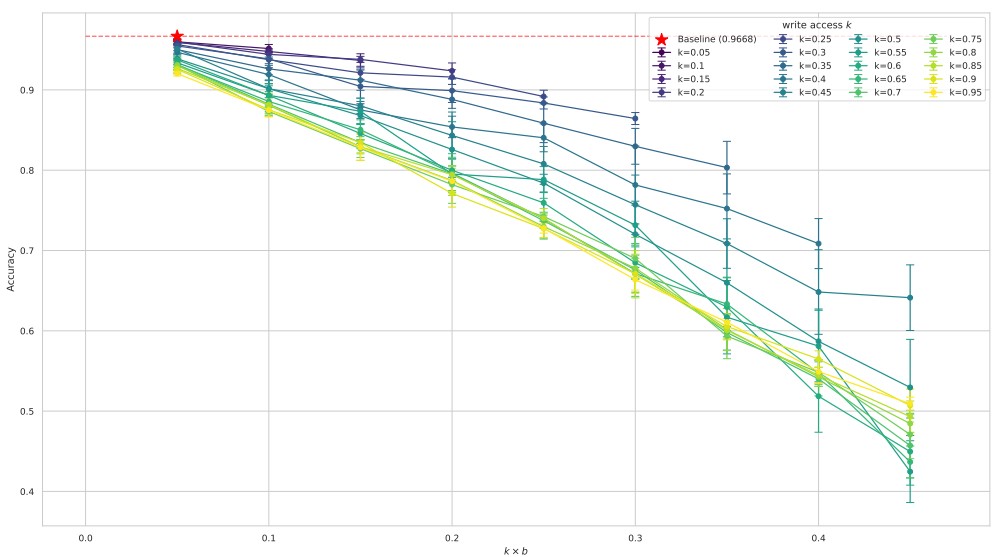

Figure 12: 2-layer vanilla MLP model on MNIST10 dataset

The curves for different write-access values $k$ remain well separated, confirming the same trend seen in the logistic regression experiments: write-access is more important than the local flipping budget. The MNIST10 task is relatively easy, and the MLP achieves a high baseline accuracy. Yet even here, modest flipping budgets already produce visible degradation.

In contrast to MNIST10, results on CIFAR10 are noisier and the magnitude of degradation is smaller. Logistic regression is a weak model for CIFAR10, and the baseline accuracy is close to random guessing. As a result, the attack has limited room to reduce performance further, and much of the observed variance comes from the underlying difficulty of the task (maybe related to the signal to noise ratio of the data) rather than the attack.

Moving from logistic regression to a 2-layer MLP increases representational capacity, and the attack again becomes effective. While the curves remain relatively noisy, the overall trend is that high k settings usually incur the most damage. This demonstrates that the greedy rule generalizes reasonably well to non-convex models, even though we do not claim theoretical optimality beyond logistic regression.

CIFAR100 is very challenging for linear models, which explains the very low ($\approx 6\check{}7\%$) baseline accuracy. Despite this low ceiling, our attack still reduces accuracy in most settings. The degradation is smaller in

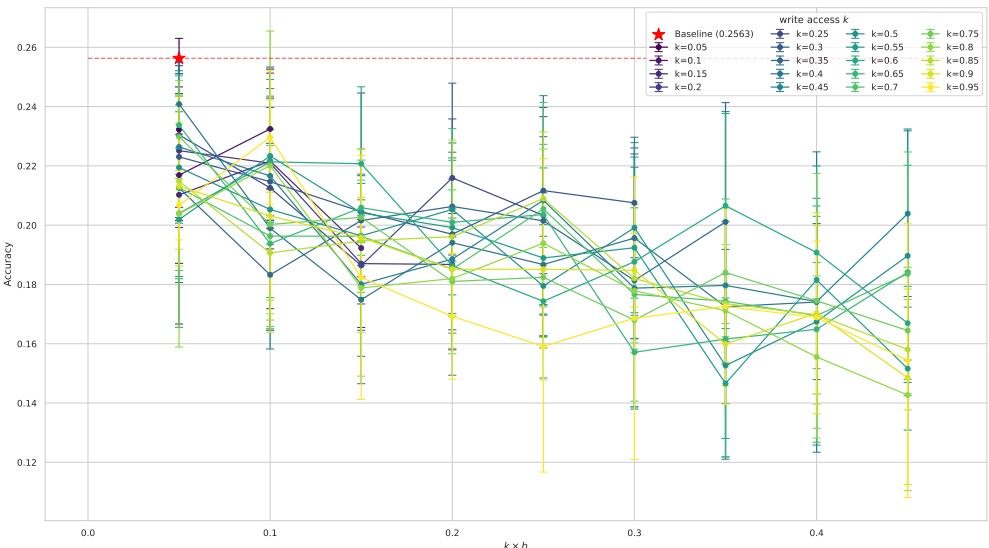

Figure 13: Logistic regression model on CIFAR10 dataset

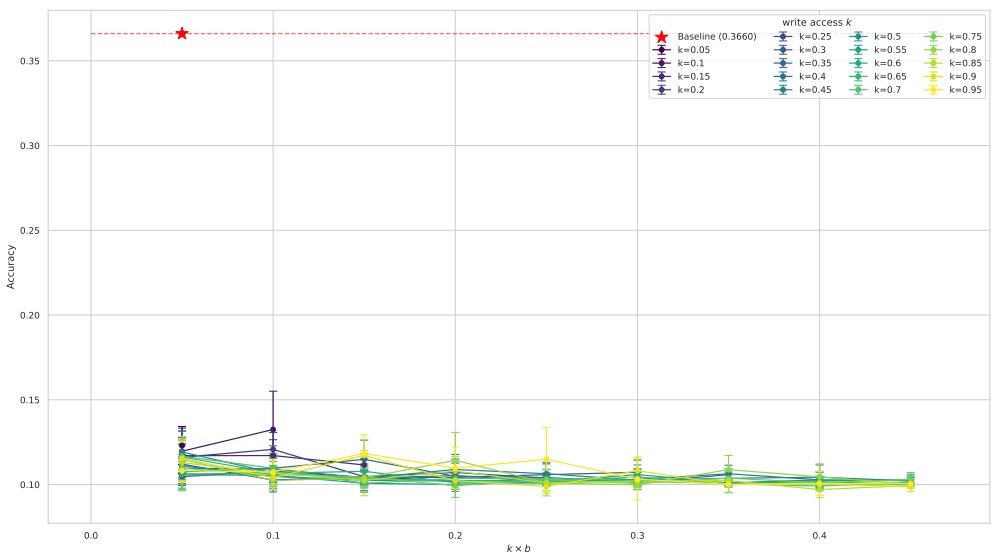

Figure 14: 2-layer vanilla MLP model on CIFAR10 dataset

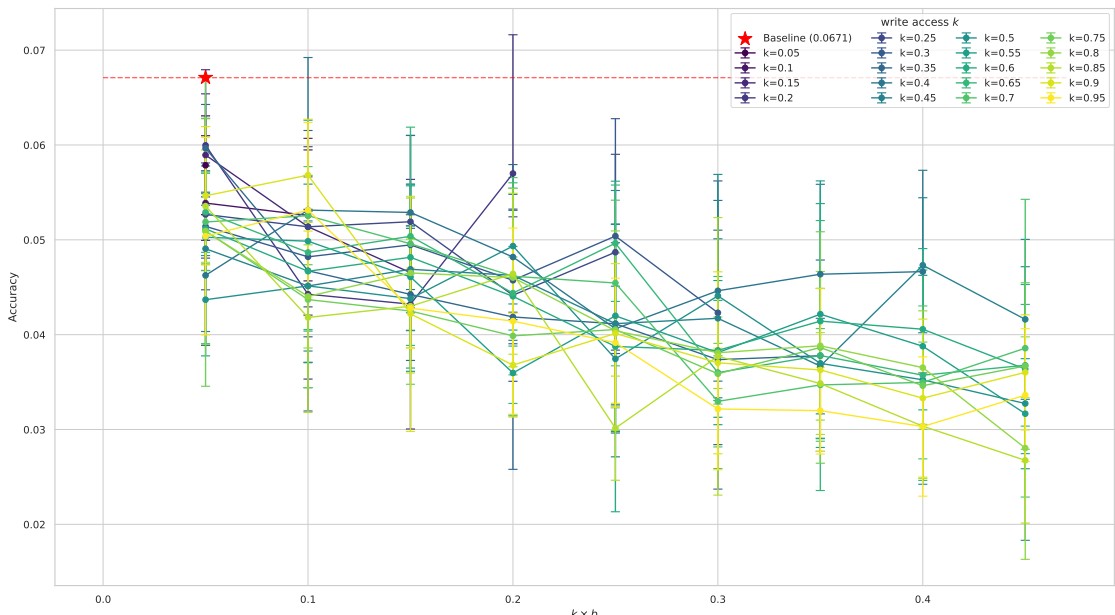

Figure 15: Logistic regression model on CIFAR100 dataset

absolute terms—again because there is little usable accuracy to destroy—but the trend across k remains visible: large write-access allows the attacker to consistently push gradients in harmful directions, even in high-class-count regimes.

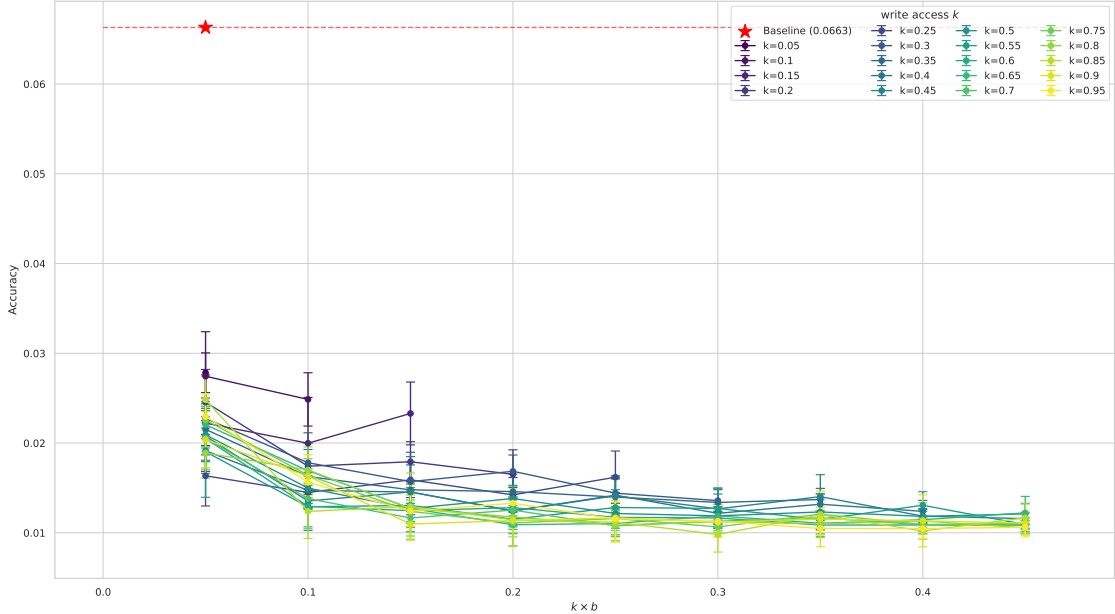

Figure 16: 2-layer vanilla MLP model on CIFAR100 dataset

As seen with CIFAR10, the attack again becomes more harmful than with a logistic regression. The same trend persists: generally, high write-access ($k$) settings produce the most damage, and the attack is efficient.

The following figure presents preliminary results demonstrating that our attack remains effective even on substantially larger and more expressive architectures. In particular, we evaluate the untargeted attack

against a Vision Transformer trained on CIFAR-10. The observed accuracy degradation across different attacker budgets indicates that the vulnerability we study is not confined to linear or shallow models.

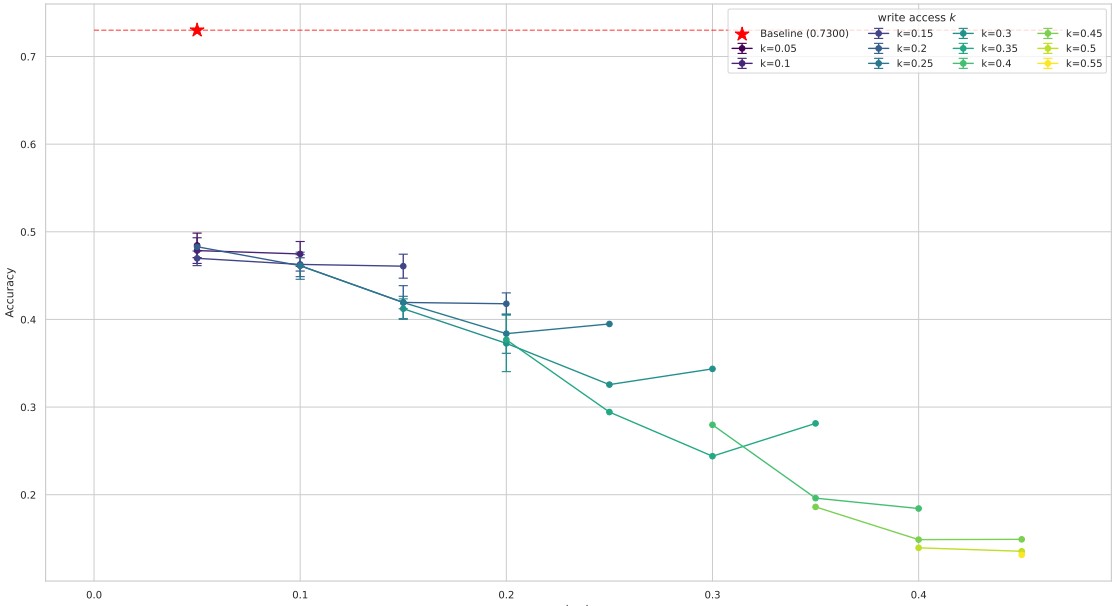

Figure 17: Vison Transformer on CIFAR10 dataset

# D    Additional Experiments on Performance Metrics and Model Variations

We provide additional results on the robustness of the models under different performance metrics and optimization strategies for the logisitc regression on the Mnist dataset.

Figure 18 reports the F1 score as a function of the corrupted fraction $k \times b$. Similar to the test accuracy trends, the F1 score decreases steadily as the corruption level increases.

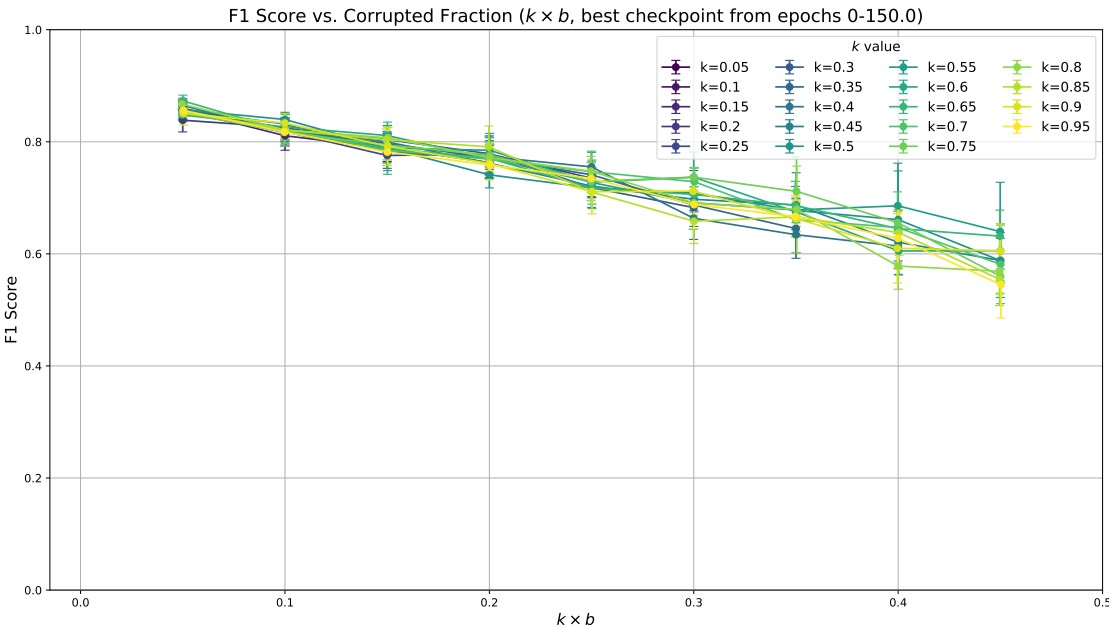

Figure 18: F1 score as a function of $k \times b$ with early stopping and an SGD optimizer.

So far, all results have used SGD optimization. We now contrast these findings with the Adam optimizer. Figure 19 seems to suggest that the attack is more effective with Adam compared to SGD.

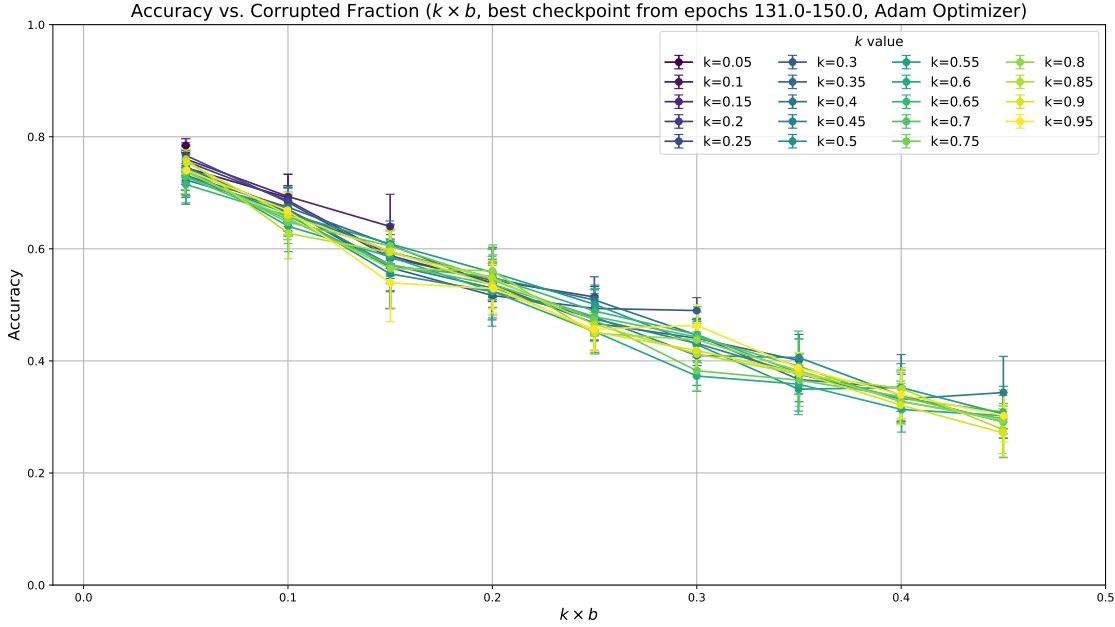

Figure 19: Adam optimizer: accuracy versus $k \times b$

# E Detectability and data normalization.

In this section, we evaluate how detectable our label-flipping attack is under common feature-preprocessing pipelines, focusing on the gradient-norm statistics that many FL defenses monitor. Practical FL systems

Table 3: Normalized-data mean gradient-norm ratio (attacker mean gradient norm divided by honest mean gradient norm). Ratios extremely close to 1 indicate that, under standard dataset normalization, the attacker's per-round gradient magnitudes are statistically indistinguishable from those of honest clients.

| dataset | model | attack/honest norm ratio |
|---------|-------|--------------------------|
| cifar10 | logreg | 1.004523 |
| cifar10 | vanilla | 1.000400 |
| mnist10 | logreg | 1.028216 |
| mnist10 | vanilla | 1.018608 |

Table 4: Unnormalized-data mean gradient-norm ratio. Without input normalization, the attacker's gradient norms become significantly inflated, making simple norm-based defenses (e.g. clipping, outlier filtering) much more likely to detect the malicious client.

| dataset | model | attack/honest norm ratio |
|---------|-------|--------------------------|
| cifar10 | logreg | 1.319266 |
| cifar10 | vanilla | 1.333890 |
| mnist10 | logreg | 7.306702 |
| mnist10 | vanilla | 232.278825 |

almost always apply some form of input normalization or scaling before training. Under these normalized settings, we empirically find that the attacker's per-client gradient norms become nearly indistinguishable from those of honest clients (Table 3). As a result, defenses that rely solely on gradient-norm thresholds or magnitude-based outlier filtering are ineffective. In contrast, when data are left unnormalized, the attacker's gradient norms may inflate (Table 4) which would make norm-based detection almost trivially successful.

## F    Accuracy evolution under the untargeted attack

The following plots illustrate how the model's accuracy evolves during training under an *untargeted* label-flipping attack on a logistic regression model for various corrupted fractions $k \times b$, where $k$ represents the write-access and $b$ is the flipping budget, for different combinations of $k$ and $b$. Each figure corresponds to a single corrupted fraction and is divided into two subplots for clearer visualization. The stability of the accuracy after long training shows that the label flipping attack is able of genuine harm, and isn't just a slowdown attack.

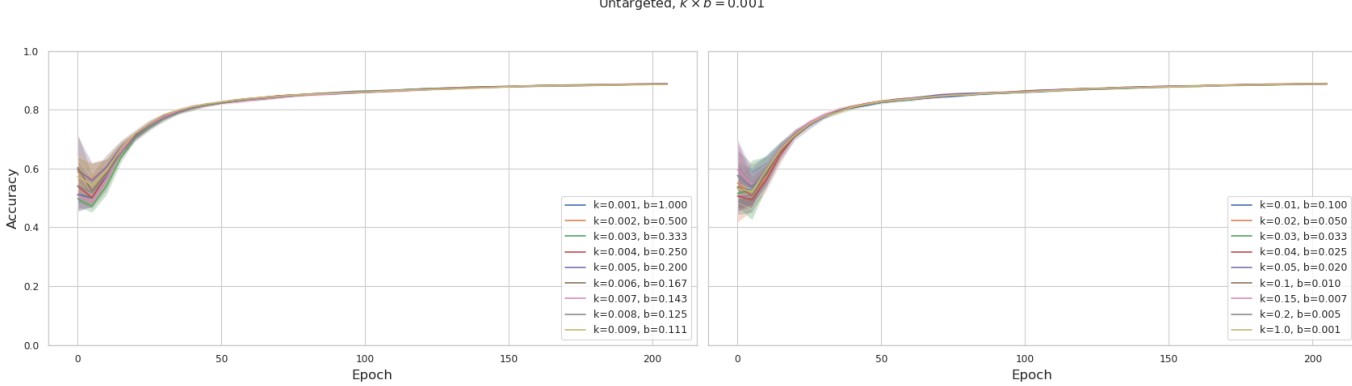

Figure 20: (Untargeted Attack) Evolution of model accuracy over training epochs with a corruption fraction of $k \times b = 0.001$, where $k$ represents the write-access and $b$ is the flipping budget, for different combinations of $k$ and $b$.

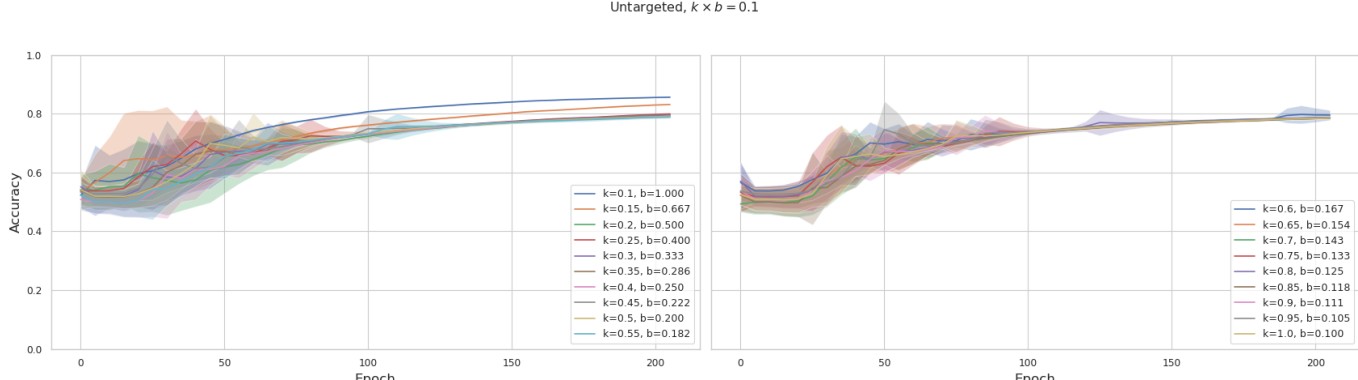

Figure 21: (Untargeted Attack) Evolution of model accuracy over training epochs with a corruption fraction of $k \times b = 0.1$, where $k$ represents the write-access and $b$ is the flipping budget, for different combinations of $k$ and $b$.

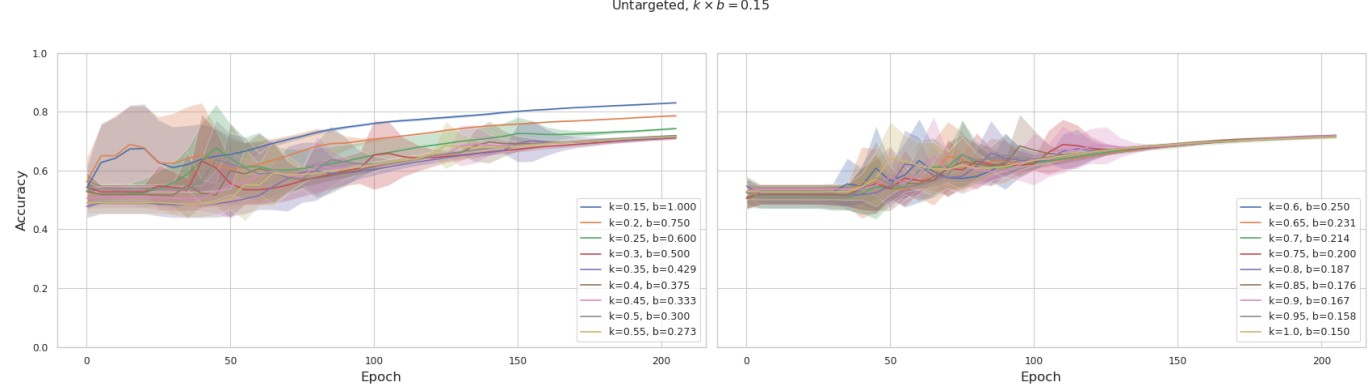

Figure 22: (Untargeted Attack) Evolution of model accuracy over training epochs with a corruption fraction of $k \times b = 0.15$, where $k$ represents the write-access and $b$ is the flipping budget, for different combinations of $k$ and $b$.

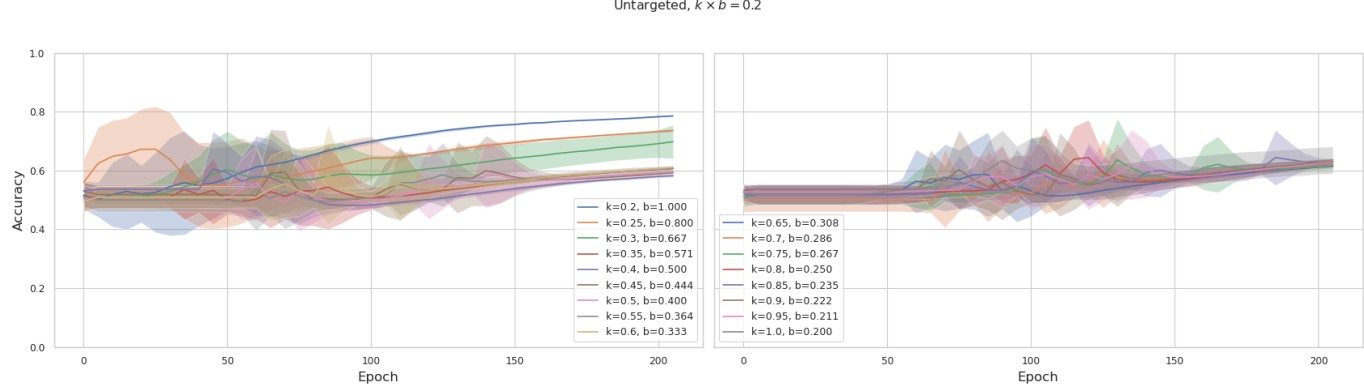

Figure 23: (Untargeted Attack) Evolution of model accuracy over training epochs with a corruption fraction of $k \times b = 0.2$, where $k$ represents the write-access and $b$ is the flipping budget, for different combinations of $k$ and $b$.

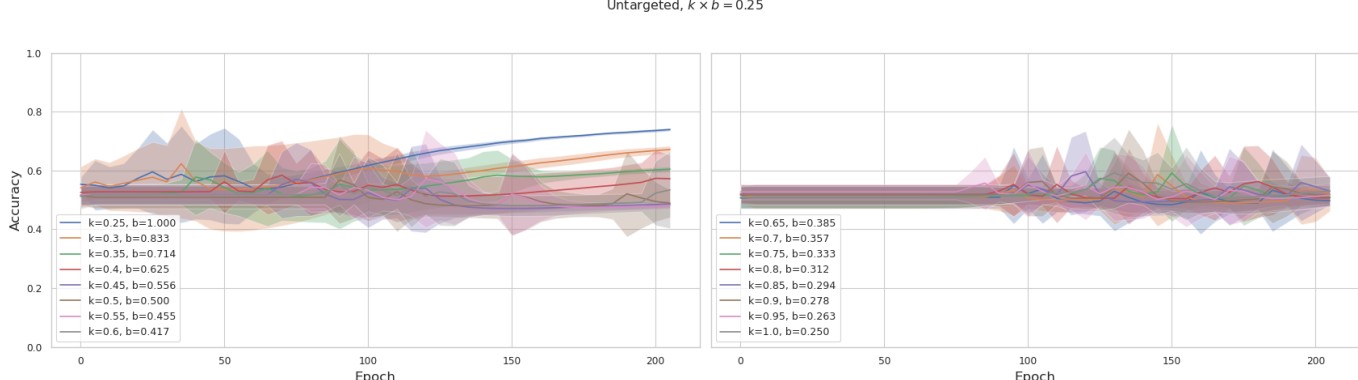

Figure 24: (Untargeted Attack) Evolution of model accuracy over training epochs with a corruption fraction of $k \times b = 0.25$, where $k$ represents the write-access and $b$ is the flipping budget, for different combinations of $k$ and $b$.

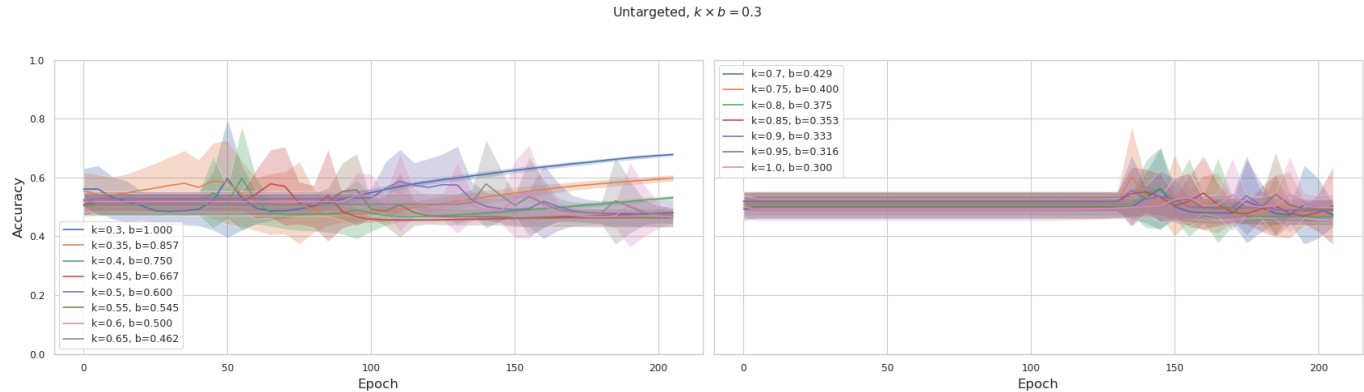

Figure 25: (Untargeted Attack) Evolution of model accuracy over training epochs with a corruption fraction of $k \times b = 0.3$, where $k$ represents the write-access and $b$ is the flipping budget, for different combinations of $k$ and $b$.

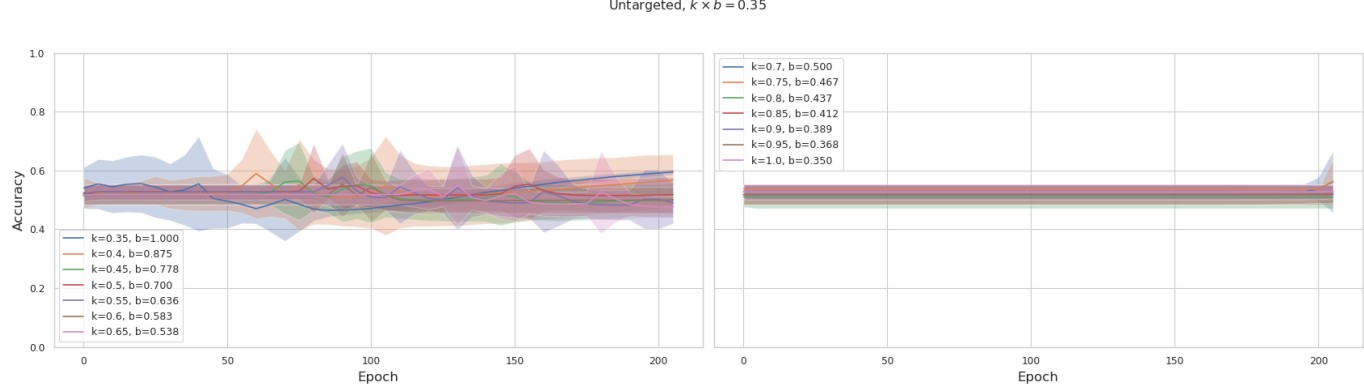

Figure 26: (Untargeted Attack) Evolution of model accuracy over training epochs with a corruption fraction of $k \times b = 0.35$, where $k$ represents the write-access and $b$ is the flipping budget, for different combinations of $k$ and $b$.

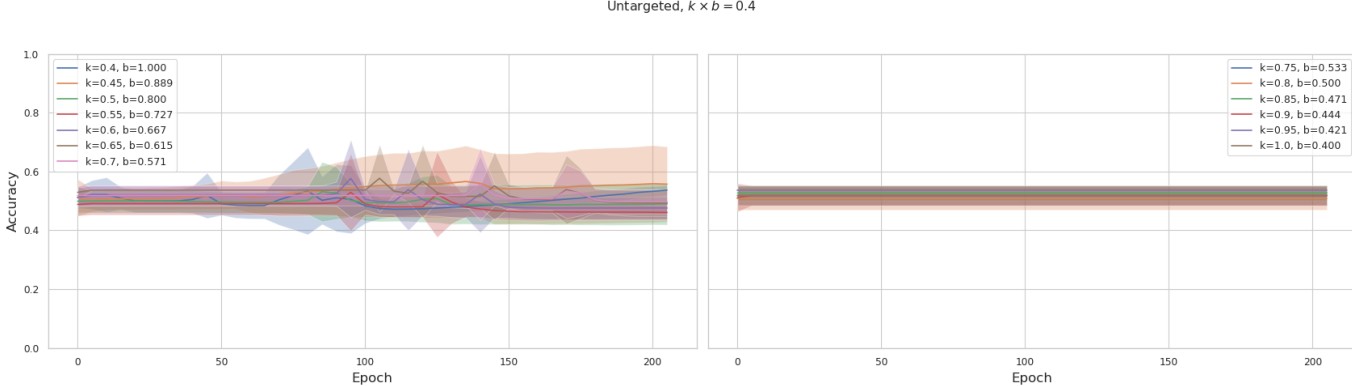

Figure 27: (Untargeted Attack) Evolution of model accuracy over training epochs with a corruption fraction of $k \times b = 0.4$, where $k$ represents the write-access and $b$ is the flipping budget, for different combinations of $k$ and $b$.

