# OpenReview forum: "Approaching the Harm of Gradient Attacks While Only Flipping Labels"
_TMLR — Rejected by TMLR_

### Review · Reviewer_xj59 · 2026-02-16

**Summary Of Contributions:**

This paper studies the label-flipping-based attacks against federated learning. The attackers modify a portion of the local training data in each training round to reduce the global model's test accuracy. The authors formalize this attack on the binary logistic regression model and propose a greedy algorithm that is proven to be optimal. Experiments on MNIST, CIFAR-10, and CIFAR-100 show that it can reduce the model performance.

Strengths:
1. It studies an important security problem about usability attacks in FL.
2. It formalizes the problem and proves the optimality of the greedy algorithm.
3. Evaluation on three datasets shows its effectiveness.

Weaknesses:
1. Only one FL setting based on the average aggregation strategy was evaluated.
2. This attack can work for the horizontal FL where the attacker knows the ground truth labels.
3. No defenses in addition to the norm-based detection are evaluated.
4. The norm-based detection can easily distinguish the attacker.
5. It's unclear how many clients there are in total, and how they are selected in each round.
6. The budget is misleading because a different set of labels can be changed in each iteration. The amount of the actually modified labels exceeds the defined budget.

**Audience:**

Yes

**Audience Explanation:**

This paper addresses an important security problem, so it's interesting to some individuals.

**Broader Impact Concerns:**

It doesn't have such a statement. It would be better to address the negative impact of the proposed attack.

**Claims And Evidence:**

No

**Claims Explanation:**

Please refer to the weaknesses. The evaluation didn't provide enough evidence for the generalizability of this method and its robustness against defenses.

**Requested Changes:**

Some changes are necessary to address the weaknesses.

1. Only one FL setting based on the average aggregation strategy was evaluated. Although the authors justified the setting in large-scale development, it's not very convincing. This is related to the 5th point. If there are "thousands to billions of clients", a very small portion of poisoned data won't affect too much.

2. This attack can work for the horizontal FL where the attacker knows the ground truth labels. This is a limitation that may not be easily addressed. Pherphas, add it to the limitation.

3. No defenses in addition to the norm-based detection are evaluated. This is very important because if a fragile attack cannot pose a real threat.

4. The norm-based detection can easily distinguish the attacker. The attack will produce a significantly different signal, making the malicious client an outlier and easy to exclude.

5. It's unclear how many clients there are in total, and how they are selected in each round.

6. The budget is misleading because a different set of labels can be changed in each iteration. The amount of the actually modified labels exceeds the defined budget.

---

> ### Author Response · Authors · 2026-03-04
>
> We very warmly thank the reviewer for careful and constructive feedback. Below we reply point-by-point to the major concerns.
>
> Please also note that some key changes in the manuscript and appendix are highlighted (colored), **including a change in the title** to better incorporate some concerns from reviewers about match between the old title of the paper and the obtained results. We also added a broader impact statement as kindly suggested by the reviewer.
>
> We emphasize that mean aggregation remains the de facto baseline in both federated learning research and large-scale deployments. This is not simply a 'naive' choice - it is favored for its scalability, communication efficiency, and ease of implementation [3]. As noted in [4], production cross-device FL systems with thousands to billions of clients often employ plain averaging, and empirical evidence suggests that even the most basic, non-robust FL algorithms can be surprisingly resilient in realistic, low-compromise settings. In this context, showing that a constrained, label-flipping adversary can meaningfully degrade model performance under mean aggregation is already a strong and practically relevant finding, given the current dominant belief in the FL community, especially from close-to-production teams as in [4].
>
> Our focus on mean aggregation is therefore **intentional**: it establishes a baseline benchmark for the impact of label-flipping attacks. If such an attack fails here, it is unlikely to succeed against more sophisticated robust aggregators; if it succeeds, it provides a clear lower bound that any robust method should surpass. Moreover, our attack design is specifically derived for the mean operator, and different aggregation rules would require different attack formulations. In real-world deployments, where FedAvg is still prevalent, understanding vulnerabilities under mean aggregation is crucial for designing defenses.
>
> Further, our results are meant as an opening for constrained label flipping attacks. By demonstrating that the constrained label-flipping attack is feasible against the most common aggregation rule, we establish the case for extending our methodology to robust aggregation methods as a natural next step. We also note that some defenses like norm clipping (which are shown very efficient in practice in combination with mean aggregation [2]) are inherently blind against our approach, since the attack operates in the space of valid gradient updates derived from valid gradient vectors generated by plausible label flips, where feature inputs are untouched. Since every flipped label still produces a legitimate gradient, clipping based solely on gradient norm is unable to distinguish or suppress malicious updates that conform to honest-looking magnitudes-making our attack inherently robust against such defenses. This further values studying mean aggregation as a baseline before moving to more complex settings.
>
> 1. As detailed in our revised Section 2.2, our setup abstracts the learning process by treating each data point as an independent Gradient Generation Unit (GGU). Conceptually, this abstraction loosely mirrors the cybersecurity principle of compartmentalization: isolating local components and their computations prior to their interaction with the central aggregator.
>
>      We agree with the reviewer that if a different, non-linear aggregator were employed, the  specific configuration of data points among clients would heavily impact the attack's efficacy. However, under standard mean aggregation, the mathematical outcome of the global update remains exactly equivalent regardless of how IID data points are partitioned or grouped into distinct client nodes.
>
>      Also, our choice of an IID setup was deliberate, intended to evaluate the adversary under the most restrictive conditions possible. As established in prior literature [7, 8], data heterogeneity generally aids poisoning efforts, whereas an IID setup acts *a priori* as a natural disadvantage for the attacker. The central message of our evaluation is that our constrained label-flipping attack succeeds in inducing model degradation *even* in these highly unfavorable IID conditions.
>
>
>
> 2. Thank you for this remark. Indeed, our attack assumes the adversary has access to the ground-truth labels. We have added this to the section 2.2 in the revised manuscript.

---

> ### Author Response · Authors · 2026-03-04
>
> 3. Evaluating a comprehensive suite of defenses is not the goal of this paper. Our defense experiments serve to establish a baseline and to show that the attack is not trivially detectable. Gradient norm detection is widely used alongside other defenses primarily because it is low-cost and computationally efficient.
>
>     As detailed in the results, **when data is normalized**, which is a common preprocessing step in production pipelines, our attack **successfully bypasses the gradient norm defense**. By demonstrating that this attack evades a common production baseline under standard scenarios, we focus on the relevance of the label-flipping threat. We do acknowledge that if the data is not normalized, the attack fails to bypass norm-based detection, and we have discussed this in the section 6 (paragraph: Detectability and data normalization) and Appendix F.
>
>
> 6. In the manuscript, we defined the budget as a per-epoch budget, rather than a cumulative budget. We included it in the notation table to be clearer.
>      To contextualize this in a real-world scenario: imagine a federated learning application where the features are user content (e.g. social media posts, ads) and the labels are a finite number of user reactions (e.g. "like" or "dislike").
>      An attacker might control a finite set of malicious accounts or compromised devices (e.g., an Astroturfing campaign). Their ability to flood the system with fake reactions during any single training round is physically, computationally, or financially limited (this limit represents our per-iteration budget). While the attacker might target a different set of posts in the next iteration, they can never exceed their maximum capability (budget) at any single point in time.
>     Furthermore, this per-iteration constraint aligns with the necessity of attacker stealthiness. Even if an adversary possesses a high level of access and the cumulative capacity to manipulate a large portion of the dataset, altering a massive fraction of labels simultaneously would easily trigger standard anomaly detection systems. By deliberately restricting their interventions to a strict, low budget per iteration, and shifting their targeted samples across different epochs, the attacker tries to feign benign noise. Thus, the per-iteration budget could be a strategic necessity for a stealthy adversary aiming to steadily degrade model availability without being expelled from the federated system.
>
> We hope to have addressed all your concerns. We remain at your disposal may you have any further questions or require additional information.
>
>
> [1] Chen, Yudong, Lili Su, and Jiaming Xu. "Distributed statistical machine learning in adversarial settings: Byzantine gradient descent." Proceedings of the ACM on Measurement and Analysis of Computing Systems 1.2 (2017): 1-25.
>
> [2] Ziteng Sun, Peter Kairouz, Ananda Theertha Suresh, H. Brendan McMahan. "Can You Really Backdoor Federated Learning?" arXiv preprint arXiv:1911.07963 (2019).
>
> [3] McMahan, H. Brendan, Eider Moore, Daniel Ramage, Seth Hampson, and Blaise Agüera y Arcas. "Communication-Efficient Learning of Deep Networks from Decentralized Data." Proceedings of the 20th International Conference on Artificial Intelligence and Statistics (AISTATS), 2017. arXiv preprint arXiv:1602.05629 (2016).
>
> [4] Shejwalkar, Virat, Amir Houmansadr, Peter Kairouz, and Daniel Ramage. "Back to the Drawing Board: A Critical Evaluation of Poisoning Attacks on Production Federated Learning." IEEE Symposium on Security & Privacy (Oakland), 2022. https://arxiv.org/abs/2108.10241 .
>
> [5] El Mahdi El Mhamdi, Rachid Guerraoui, Sébastien Rouault. The Hidden Vulnerability of Distributed Learning in Byzantium. ICML 2018 Available at: https://arxiv.org/abs/1802.07927
>
> [6] Cong Xie, Oluwasanmi Koyejo, Indranil Gupta. Generalized Byzantine-tolerant SGD. Available at: https://arxiv.org/abs/1802.10116
>
> [7] El-Mahdi El-Mhamdi, Sadegh Farhadkhani, Rachid Guerraoui, Arsany Guirguis, Lê Nguyên Hoang, Sébastien Rouault. Collaborative Learning in the Jungle (Decentralized, Byzantine, Heterogeneous, Asynchronous and Nonconvex Learning). Available at: https://arxiv.org/abs/2008.00742
>
> [8] Sai Praneeth Karimireddy, Lie He, Martin Jaggi. Byzantine-Robust Learning on Heterogeneous Datasets via Bucketing. Available at: https://arxiv.org/pdf/2006.09365

---

### Review · Reviewer_hV3i · 2026-02-20

**Summary Of Contributions:**

The paper studies availability attacks in federated learning under a constrained threat model where attackers can only flip labels. The authors define a label flipping attack against logistic regression when the server uses mean aggregation as a budget-constrained optimization problem. The attacker is assumed omniscient and adapts label flips at every training epoch. Then, they derive a greedy algorithm claimed to be per-epoch optimal for both untargeted and targeted attacks. This framework is also extended to multi-class linear classification. Experiments on MNIST, CIFAR-10, and CIFAR-100 show that even small label-flipping budgets can reduce test accuracy under mean aggregation. The paper also studies the trade-off between write-access and local flipping budget.

Strengths:
+ The paper is generally well organized and aims to address a clearly stated research question. The idea of studying more constrained adversaries in federated learning settings is conceptually interesting and less explored in the research literature.

+ The experimental analysis showing that write-access (k) is more impactful than local budget (b) is interesting and potentially useful for understanding poisoning attacks behavior.

+ The detectability analysis under data normalization in Appendix F provides a useful practical observation: label flipping attacks can evade norm clipping defense strategies.


Weaknesses:
+ The threat model is not well motivated from a practical standpoint: The paper simultaneously assumes an omniscient attacker (with full access to model parameters and all participants’ data at every round) and restricts the attacker’s capabilities to flipping labels only. These two assumptions are incompatible in practice: any omniscient adversary would have no reason to limit themselves to label flipping. They could perform model poisoning or direct gradient manipulation, rendering more powerful attacks. On the other hand, realistic label-flipping attackers are those who corrupt the training data before the training begins (e.g., compromised data gathering, malicious crowdsourcing workers during the annotation process), and such attackers are not omniscient.

+ The attacker’s objective is not well formulated: the untargeted attack objective aims to anti-align the poisoned gradient with the aggregated gradient from the honest participants. When honest points substantially outnumber the poisoning points (which is the case under the small budgets that the paper considers and highlights), this attack strategy merely reduces the magnitude of the net update without reversing it. In other words, the model still converges in the correct direction, but more slowly. Thus, this is not really an availability attack that aims at maximizing the model’s error. For that, the attacker’s objective should be reconsidered. The same issue affects the targeted attack, where convergence to the target model is not guaranteed under small budgets. This flaw undermines the theoretical core of the paper.

+ Reduced scope: the paper focuses on linear classification and in the experiments, just a shallow MLP is tested. The paper considers very restrictive federated learning scenarios: IID datasets for the participants, only mean aggregation is considered.

+ Following the previous point, the experimental evaluation does not reflect a genuine federated learning setting: The paper evaluates only logistic regression under IID conditions with mean aggregation, which is the setting most favorable to the attack and least representative of real deployments. The paper never states the number of participants, suggesting that the experiments are conducted in a centralized setting, just corrupting some batches of training data. In this sense, there is no exploration of how the attacks perform against a different number of clients, varying the fraction of compromised clients, evaluating non-IID scenarios, or testing against federated averaging rather than simple mean aggregation, just to cite some.

+ Lack of comparison with other methods: There is no comparison with other label flipping strategies, making it impossible to assess how much the greedy optimality actually contributes over naïve random flipping or other strategies used for data poisoning in practice. There is no evaluation against robust aggregation methods, which is a standard practice for testing poisoning attacks in federated learning. Similarly, there is no comparison with other data or model poisoning attack strategies in federated learning. In this sense, the paper’s claim of “approaching the harm of gradient attacks” is never empirically substantiated. Experiments on deep architectures typically used in practical applications (CNN or larger DNNs) are not considered.

**Audience:**

Yes

**Audience Explanation:**

The exploration of more restrictive attacker settings in federated learning is an interesting problem and, as mentioned before, there are some results that can be of interest for TMLR’s audience, such as the experimental analysis showing that write-access can be more impactful than local budget or the detectability analysis under data normalization provided in Appendix F which shows that label flipping can help evading norm-clipping-based defense strategies.

**Claims And Evidence:**

No

**Claims Explanation:**

The paper makes three main claims: (1) the greedy algorithm is optimal at each training step; (2) label flipping can approach the harm of gradient attacks; (3) small budgets suffice for availability attacks. However, none of these are well supported.

Claim (1) is mathematically correct, but it is undermined by the flawed objective (as explained before). In this sense, the algorithm is optimal with respect to a proxy that does not correctly capture the attacker’s true goal (e.g., maximize the overall error in the case of the untargeted attack). Thus, optimality with respect to a proxy that is not necessarily representative of the true objective is not a meaningful guarantee.

Claim (2) is not supported empirically. There is no direct comparison between the proposed attack and gradient-based attacks under equivalent conditions. Without this, the title and the abstract promise something that the experiments never support.

Claim (3) is also not well supported. The paper presents accuracy drops under small budgets as evidence of a successful availability attack, but it never establishes that the observed degradation is caused by genuine model corruption rather than simple training slowdown due to effective norm-gradient reduction. As discussed earlier and given the flaw in the objective formulation, this is the more plausible explanation for small budget results.

**Requested Changes:**

+ Justification and validity of the threat model: How is it reasonable that an omniscient attacker that can intervene at each training epoch is restricted to label flipping or chooses to use a label flipping strategy instead of model poisoning?

+ Attacker’s objective: The correct untargeted objective should account for the full aggregated gradient and aim to maximize the loss on the entire dataset (or the dataset from the malicious participant) as a proxy to maximize the overall error. In this sense, the authors should either reformulate accordingly or justify formally why their proxy objective is a valid approximation with theoretical and empirical support.

+ Provie empirical comparison with gradient-based attacks under equivalent conditions. Without this, the central claim of the paper cannot be evaluated.

+ Experiments in more realistic settings: federated averaging and robust aggregation techniques, non-IID scenarios, more advanced CNN/DNN architectures, varying number of participants and the fraction of malicious participants. Compare to other random label flipping strategies.

+ Disentangle the training slowdown from genuine availability attack as discussed earlier. For this, the paper should measure not just final accuracy but show training trajectories or convergence rates to demonstrate that the attack causes the true model degradation rather than slower convergence that eventually recovers.

+ Remove the omniscient assumption and replace it with a more realistic partial-knowledge threat model where, for example, the attackers don’t have access to the whole training set (from all participants) but only to the data from the malicious participants.

---

> ### Author Response · Authors · 2026-03-04
>
> We thank the reviewer for their thorough feedback on our work. We appreciate them acknowledging the paper’s novelty and finding its technical depth compelling.
>
> Please also note that some key changes in the manuscript and appendix are highlighted (colored), **including a change in the title** to better incorporate some concerns from reviewers about match between the old title of the paper and the obtained results.
>
> ### 1. Threat model and omniscience
>
> Our adoption of an omniscient attacker is mostly useful for **comparability** (in terms of writing power on the model) with prior work on gradient attacks (Byzantine/omniscient threat models) that are relevent in the distributed/federated learning context (this also influenced the change in the title).
> As argued by [4], an “apple-to-apple” comparison between data poisoning and gradient attacks is not straightforward because the literature typically studies them under different threat models with different levels of knowledge and intervention capabilities. To remove this confounding factor, we follow [4] and consider a threat model in which the attacker is omniscient and can intervene at every iteration as in [1, 2, 3, 4].
>
> In Section C.5, we also discuss how a limited-knowledge attacker can operate using a local surrogate; however, this is not the main goal of our work.
>
>
> To contextualize the attack in a real-world scenario: imagine a federated learning application where the features are user content (e.g. social media posts, ads) and the labels are a finite number of user reactions (e.g. "like" or "dislike").
>
> An attacker might control a finite set of malicious accounts or compromised devices (e.g., an Astroturfing campaign). Their ability to flood the system with fake reactions during any single training round is physically, computationally, or financially limited (this limit represents our per-iteration budget). While the attacker might target a different set of posts in the next iteration, they can never exceed their maximum capability (budget) at any single point in time.
>
> Furthermore, this per-iteration constraint aligns directly with the necessity of **attacker stealthiness**. Even if an adversary possesses a high level of access and the cumulative capacity to manipulate a large portion of the dataset, altering a massive fraction of labels simultaneously would easily trigger standard anomaly detection systems. By deliberately restricting their interventions to a strict, low budget per iteration, and shifting their targeted samples across different epochs, the attacker tries to feign benign noise. Thus, the per-iteration budget could be a necessity for a stealthy adversary aiming to steadily degrade model availability without being expelled from the federated system.
>
>
> ### 2. Objective formulation and harm
>
> The first problem (Equation 1) accounts for the full aggregated gradient: it maximizes the misalignment between the gradient of the poisoned full dataset and the gradient that would be obtained from the clean dataset. However, as shown in Section 3.1, since the aggregation rule is linear and the attacker optimizes over their write access (i.e. the subset of labels they can modify under the budget constraint), this reduces to Equation 3. The resulting objective depends only on the portion of the gradient controlled by the attacker and is therefore the correct per-iteration objective under linear aggregation.
>
> The plots in Appendix G illustrate how the model’s accuracy evolves during training under an untargeted label-flipping attack on logistic regression for various corrupted fractions $k×b$, where $k$ denotes write access and $b$ the flipping budget. Each figure corresponds to a fixed corrupted fraction and contains two subplots for clarity. The fact that the loss and accuracy stabilize after sufficient training indicates that the attack causes genuine degradation rather than merely slowing convergence.

---

> ### Author Response · Authors · 2026-03-04
>
> **Regarding the comparison with random label flipping**: we prove that our algorithm is the optimal greedy label-flipping strategy for the considered objective. Random label flipping can be viewed as a greedy strategy in the sense that flips are performed independently at each iteration without global coordination. Since our method is provably optimal among greedy strategies under the same budget constraint, it is theoretically stronger than random label flipping for the same setting.
>
>
> ### 3. On the comparison with gradient based attacks
>
> Our goal is to quantify the extent of harm achievable through label flipping alone. In response to the reviewer’s concern, we have moderated the tone of our claims regarding gradient-based attacks in both the introduction and abstract. Our intended message was that label flipping can **initiate** an availability attack and therefore, in this sense, *approaches* the harm of gradient-based attacks.
>
> ### 4. Model depth
>
> Beyond logistic regression, we evaluated a MLP and conducted preliminary experiments with a Vision Transformer (ViT). These additional results, reported in Appendix D, indicate that the attack remains effective on deeper architectures.
>
>
> We hope to have addressed all your concerns. We remain at your disposal may you have any further questions or require additional information.
>
> [1] Peva Blanchard, El Mahdi El Mhamdi, Rachid Guerraoui, and Julien Stainer. Machine learning with adversaries: Byzantine tolerant gradient descent. Advances in neural information processing systems, 30,
> 2017.
>
> [2] El Mahdi El Mhamdi, Rachid Guerraoui, Sébastien Rouault. The Hidden Vulnerability of Distributed Learning in Byzantium. ICML 2018 Available at: https://arxiv.org/abs/1802.07927
>
> [3] Gilad Baruch, Moran Baruch, and Yoav Goldberg. A little is enough: Circumventing defenses for distributed learning. Advances in Neural Information Processing Systems, 32, 2019.
>
> [4] Bouaziz, W., Usunier, N., & El-Mhamdi, E. (2024). Inverting Gradient Attacks Makes Powerful Data Poisoning. Trans. Mach. Learn. Res., 2025.

---

### Review · Reviewer_4vuM · 2026-02-25

**Summary Of Contributions:**

This paper investigates constrained label-flipping availability attacks on federated learning with mean aggregation, making key contributions: it formalizes the attack on logistic regression as a budget-constrained optimization problem and derives a per-training-step optimal greedy algorithm; validates the attack’s potency empirically (0.1% label flips cut accuracy by 6%, 25% flips render models near random guessing); uncovers that an attacker’s write-access ratio (k) is more impactful than local flipping budget (b); defines and compares targeted/untargeted attack variants; extends the attack framework to multi-class tasks with a generalized optimal algorithm; and shows normalization makes the attack’s gradient norms undetectable by norm-based defenses.

Strengths:
Rigorous theoretical proofs for algorithm optimality; comprehensive empirical validation on standard datasets/models with statistical reliability; novel insights into k/b trade-offs and attack variance; focus on industry-dominant mean aggregation ensures practical relevance.

Weaknesses:
The greedy algorithm is only per-step (not global) optimal; the theory is convex-model-based, with merely preliminary deep model experiments; threat model assumes an omniscient attacker (limited-knowledge case only conceptually discussed); no defense strategies proposed or tested; no analysis on robust FL aggregation mechanisms.

**Audience:**

Yes

**Audience Explanation:**

The paper’s focus on label-flipping attacks in federated learning aligns with TMLR’s scope of exploring computational/mathematical principles of learning systems and their robustness. Its findings—on low-budget attack efficacy, write-access vs. budget trade-offs, and undetectability under normalization—offer new insights into learning system behavior, which interests TMLR’s audience: FL researchers, adversarial ML experts, and security practitioners. These insights help advance understanding of practical learning system vulnerabilities, making the work relevant to those studying secure and robust learning.

**Broader Impact Concerns:**

NA.

**Claims And Evidence:**

Yes

**Claims Explanation:**

The paper’s claims are backed by rigorous theoretical proofs (per-step optimality of the greedy algorithm) and comprehensive empirical evidence—quantifiable results on MNIST/CIFAR datasets with multiple models, showing clear accuracy degradation (e.g., 0.1% label flips reduce accuracy by 6%), all with statistical reliability.

**Requested Changes:**

1. Empirical verification of attackers with limited knowledge; 2. In-depth analysis of the optimality and gradient dynamics of deep model attacks; 3. Discussion of the feasibility of a globally optimal label-flipping strategy.

---

> ### Author Response · Authors · 2026-03-04
>
> We thank the reviewer for their thorough and constructive assessment of our work. We particularly appreciate their recognition of the `theoretical rigor` and `practical relevance` of studying constrained label-flipping attacks in federated learning.
>
> Please also note that some key changes in the manuscript and appendix are highlighted (colored), **including a change in the title** to better incorporate some concerns from reviewers about match between the old title of the paper and the obtained results.
>
> Regarding the suggested additions:
>
> * While our current evaluation focuses on the standard threat model, we agree that studying limited-knowledge settings would be valuable, and we discuss this perspective in Section 7.4. However, this extends beyond the current scope, which aims to characterize the degradation power of constrained label flips under a well-defined threat model (as did [1] on poisoning with features). We view this as promising future work that goes beyond the score of our contribution (or that of [1], one of our most closely related works for instance).
>
> * Our theoretical results establish per-step optimality under convex assumptions, and our experiments include validation on deep models (Appendix C). A deeper theoretical treatment of gradient dynamics in deep networks is non-trivial and would require substantially expanding the framework. We acknowledge this as an interesting direction but consider it beyond the current contribution.
>
> * In Appendix B.2 (*On the optimality of the approach*), we discuss per-step optimal greedy optimization and clarify that global optimality is computationally intractable in general due to combinatorial complexity:
> *Constructing a globally optimal sequence of flips therefore requires solving a combinatorial, horizon-wide optimization problem (or equivalently optimizing over label sequences), which would be computationally intractable in realistic settings and would impose substantial computational overhead compared to our per-epoch rule.*
>
> We thank the reviewer once more for their valuable feedback and remain available for further clarification.
>
> [1] Bouaziz, W., Usunier, N., & El-Mhamdi, E. (2024). Inverting Gradient Attacks Makes Powerful Data Poisoning. Trans. Mach. Learn. Res., 2025.

---

### Review · Reviewer_3zjv · 2026-02-27

**Summary Of Contributions:**

**Summary**:
This paper investigates whether a constrained adversary that can only flip a small fraction of training labels can still perform effective availability attacks in federated learning with mean aggregation. The paper formulates label flipping as a per-epoch constrained optimization problem and derives a greedy algorithm that is provably optimal at each training step. They show both theoretically and empirically that even extremely small corruption rates (e.g., 0.1% of labels per epoch) can measurably degrade accuracy, while larger budgets can drive performance close to random guessing. Overall, the work demonstrates that label-only attacks can approximate the damage of stronger gradient-based attacks under realistic federated learning assumptions.

**Strengths**:
- The paper shows clear formalization and proposes an attack for a weak adversary that can only flip labels in FL settings.
- The paper demonstrates the effectiveness of the attack on benchmark datasets.

**Weakness**:
- The practicality of this threat model is questionable. The paper assumes that the adversary is weak and can only flip labels without changing features or updating gradient. However, the attack needs read-access to the parameters during training to perform the attack. Furthermore, the attack is performed per epoch, which means the adversary can even monitor the training process and flip labels at every epoch. Could you give an example of the adversary in a real-world scenario?
- The paper does not discuss other data poisoning methods in this setting. Why can the attacker only flip the labels but not change data features/update gradients? If the adversary can use other data poisoning methods, could you compare them with the proposed method, in terms of effectiveness and stealthiness?
- The term "targeted attack" seems confusing and misleading to me. Usually, by targeted attack, we mean methods that make the model return label $y_{target}$ for data with ground-truth label $y_{true}$. However, the paper only measures the test accuracy, which means the drop in the clean accuracy. Also, how can the target and untarget attacks in binary settings be different?
- The paper does not compare with baseline. What if the attack just randomly flips some samples? How effective/stealthy is that method?
- The paper only evaluates 2-layer MLP but not modern neural networks, limiting the practicality.
- The role of FL settings is not clear to me. It seems the attack should also work in standard training settings.

**Audience:**

Yes

**Audience Explanation:**

The paper discusses a novel threat model for data poisoning in FL settings, which is related to TMLR audience.

**Claims And Evidence:**

No

**Claims Explanation:**

Claim 1: The practicality of threat model. The paper assumes the adversary can only flip labels but on the other hand can monitor the training process and flip labels during training, which is questionable.

Claim 2: The attacker severely degrades the model. I have a few concerns:
- Concern 1: The paper only measures the drop in the accuracy, but in data poisoning we also want the model return the target labels.
- Concern 2: The paper does not evaluate the simple random flipping baseline.
- Concern 3: The paper only evaluates with the simple gradient norm detection, but there are many other detection defenses.
- Concern 4: The paper does not evaluate modern NNs.

**Requested Changes:**

- Justify the threat model.
- Compare with other baselines, i.e., random flipping, other data poisoning in FL that perturb datapoints or gradients.
- Conduct experiments with modern NNs and other detection methods.

---

> ### Author Response · Authors · 2026-03-04
>
> We very warmly thank the reviewer for careful and constructive feedback. We appreciate the fact that the reviewer liked the `clear formalization` of the paper and the empirical demonstration of the effectiveness of the attack.
> Below we reply point-by-point to the major concerns.
>
> Please also note that some key changes in the manuscript and appendix are highlighted (colored), **including a change in the title** to better incorporate some concerns from reviewers about match between the old title of the paper and the obtained results.
>
> * Regarding your first concern, our work focuses on availability attacks, not backdoor attacks. The goal is to degrade overall model performance (e.g., reduce accuracy or increase loss). Therefore, measuring the drop in global accuracy (and F1 score in Appendix B) seems an appropriate metric for evaluating availability degradation in our setting.
> We also clarify that the attacker is assumed to have read access to the labels contributed by other workers, but is constrained by a limited write budget. This models a realistic setting where the adversary can observe the global state but can only manipulate a restricted subset of labels per iteration (see the example provided in our response to Reviewer xj59).
>
>
> * Regarding the comparison with random label flipping: we prove that our method is the optimal greedy label-flipping strategy for the considered objective. Random label flipping can be seen as a naive greedy strategy in which labels are flipped independently at each iteration without optimizing the objective. Since our method is provably optimal among greedy strategies under the same budget constraint, it is theoretically stronger than random flipping in this setting.
>
>
>
> * Evaluating a comprehensive suite of defenses is not the primary goal of this paper. Our defense experiments aim to establish a baseline and to demonstrate that the proposed attack is not trivially detectable. Gradient norm–based detection is widely used in practice because it is computationally inexpensive and easy to deploy.
> As shown in our results, *when the data is normalized* (a common preprocessing step in production pipelines) our attack *successfully bypasses gradient norm–based detection*. By demonstrating that the attack evades this widely used baseline under realistic preprocessing conditions, we highlight the practical relevance of the label-flipping threat.
> We also acknowledge that when data is not normalized, the attack becomes more detectable under norm-based defenses. This limitation is explicitly discussed in Section 6 (“Detectability and Data Normalization”) and Appendix F.
>
>
> * In addition to logistic regression, we evaluated a multilayer perceptron and conducted preliminary experiments with a Vision Transformer (ViT). These results are reported in Appendix D and show that the attack remains effective on deeper and more modern architectures.
>
> We thank again the reviewer for their time and thorough review. We hope to have answered all your concerns and questions and remain at your disposal for further clarification.

---

### Decision · Action_Editor_PiM3 · 2026-03-30

**Recommendation:** Reject

**Additional Comments:**

The recommendation is based on the reviewers' comments, the action editor's evaluation, and the authors’ response.

This submission should not be accepted in its current form due to several fundamental issues, as pointed out by the reviewers, including

- The threat model is weakly justified and lacks strong practical justification.
- The objective is not well aligned with the stated goal of maximizing model degradation. While the derivation of their objective is mathematically correct, the proposed objective optimizes the per-iteration gradient misalignment, but not the final model degradation. On the other hand, there is no theoretical justification that this surrogate objective leads to maximal availability damage, compared to other alternatives. The additional experiments (added during the rebuttal) showing the stabilization of the loss do not resolve this issue
- The key claims remain insufficiently supported (e.g., read access in this work might be even stronger than the write access).
- Lack of sufficient evaluations and comparisons (e.g., random label flipping)

Since the rebuttal and the revision are insufficient, this paper requires significant rework and another round of full reviews. I hope the reviewers’ comments can help the authors prepare a better version of this submission.

**Audience:**

Yes

**Audience Explanation:**

of general interest

**Claims And Evidence:**

No

**Claims Explanation:**

Not entirely. The threat model and some corresponding analysis require more evidence.

**Resubmission Of Major Revision:**

The authors may consider submitting a major revision at a later time.